# Rising dengue risk with increasing El Niño–Southern Oscillation amplitude and teleconnections

Yunyu Tian[1,2,12], Yiting Xu[1,12], Yilin Liang [1,2,12], Ziqin Zhou[1,2], Katie M. Susong[3,4], Yuyang Chen[5], Kishen Joshi[3,4], Amy M. Campbell[3,4], Ahyoung Lim [3,4], Qiushi Lin[6], Zixuan Ma[7], Yuanlong Wei[1,2], Yun Yang [2], Cheng Sun [2], Juan Feng [2], Qixin He [8], Zengmiao Wang[2], Bernard Cazelles [9,10], Yuanfang Guo[11], Kai Liu [1], Oliver J. Brady [3,4] ✉ & Huaiyu Tian [1,2] ✉

Global climate variability has been linked with some of the largest dengue outbreaks, including the record-breaking 2023–2024 epidemic, but the understanding of their mechanism and evidence for their association is lacking. By incorporating reported dengue cases and climate data from 57 countries across the Americas and Asia from 1980 to 2024, we unpacked the global climate teleconnection and quantifying its impact on dengue cases. We revealed that the heterogeneity in the association between global climate variability and dengue cases across regions is affected by the strength and types of global climate teleconnections with temperature and precipitation. By controlling for the heterogeneity, 63% of the variation in dengue cases can be attributed to El Niño–Southern Oscillation fluctuations, with higher values in endemic regions. The 1982–83, 1997–98, 2015–16, and 2023–24 El Niño events were estimated to have induced an additional 0.2, 1.4, 4.1, and 9.6 million dengue cases, respectively, over regular seasonal patterns. Due to human-induced warming, El Niño events and teleconnections may cause a 39.0–81.7% increase in cumulative cases in 2020–2099. Our findings quantify the association between global climate variability and dengue epidemics and caution about the potential future risk.

As one of the most climate-sensitive diseases, dengue has seen frequent outbreaks in recent decades, with a substantial proportion widely attributed to global climate variability[1–5]. Considerable efforts have been made to mitigate the spread of dengue, such as global strategies for dengue control formulated by the World Health Organization (WHO) towards achieving the Sustainable Development Goals since 2012[6,7]. However, from 2000 to 2019, globally reported dengue cases increased 10-fold from 0.5 to 5.2 million[8], with increases particularly occurring in the Americas and Asia[8]. While dengue vaccines have recently made important progress[8,9], dengue continues to pose an escalating threat to public health due to persistent vector

transmission and the lack of vaccine availability and effective mitigation policies.

Despite the widely recognised close link between global climate variability and dengue outbreaks[10–12], varied and uncertain associations between them have been observed across different regions and periods[3,13–16]. For example, in 2023 and 2024, both El Niño climate events and subsequent dengue outbreaks reached unprecedented levels on a global scale[9,17–19]. El Niño-related increases in dengue risk occurred in tropical countries such as Sri Lanka[20], Peru[21], and Ecuador[15]; however, decreases also occurred in the Solomon Islands[14]. Some studies argued that the correlation between ENSO and dengue

incidence was weak in their respective study regions and time frames[22,23] in that interannual correlations were only observed within specific periods in Thailand[16]. Those discrepancies across temporal and spatial scales underscore the need to disentangle the impact of global climate variability on dengue epidemiology[3,24]. Given the extensive body of research, we hereafter use El Niño–Southern Oscillation (ENSO) as the well-known indicator of global climate variability, among other similar indices such as the Indian Ocean Dipole (IOD) and the Madden Julian Oscillation (MJO), to investigate this question for simplification.

Manifested as sea surface temperature anomalies in the tropical Pacific Ocean, ENSO influences dengue risk through "teleconnections" – ENSO-induced atmospheric circulation such as the Walker circulation and the propagation of Rossby waves physically alter distant regional climates, thereby affecting mosquito ecology[25,26]. While the dynamical mechanisms of these teleconnections have been well established in climatology[27,28], an open question remains regarding the latter process; namely, how do teleconnections contribute to the different associations between ENSO and dengue epidemics[29]? Previous research attempted to investigate the latter process by incorporating short-term temperature and/or precipitation anomalies during ENSO events[13]. However, the mediating effects of ENSO on dengue cases through temperature and precipitation remain unclear. Assessing the teleconnections is key to elucidating the mechanism by which climate change impacts dengue transmission[30,31].

Furthermore, ENSO teleconnections with multiple climatic variables result in complex effects on dengue epidemiology in terms of mosquito ecology[32,33]. El Niño events typically cause extreme warming and drought simultaneously in many tropical countries, in which warming promotes viral replication and vector development[34,35], whereas drought can reduce mosquito habitats[36,37]. Such potential trade-offs within the transmission mechanism highlight the importance of distinguishing the effects of ENSO-driven temperature and precipitation on dengue transmission[32,38]. Unravelling these influence pathways is crucial for early detection and forecasting of dengue outbreaks, which can reduce the burden by improving local health system preparedness and implementing effective vector control measures. The complexity of these pathways, coupled with a lack of long-term and large-scale data, constrains our understanding of how ENSO alters global dengue patterns. With increasing ENSO variability in recent years and growing evidence that variability will further increase under future emission scenarios[31,39], there is an urgent need to investigate how ENSO shapes future risk of dengue[40,41].

In this study, we assembled a dataset of dengue cases reported from 57 tropical countries across the Americas and Asia, global gridded climate data, and ENSO indices from 1980 to 2024. We assessed the impact of ENSO on dengue cases through local climate by defining country-level teleconnections between ENSO indices and population-weighted temperature and precipitation. We applied a distributed lag regression model to quantify the intricate associations between ENSO teleconnections and dengue epidemics and then estimated the effects of future ENSO changes on dengue epidemics.

## Results

### Heterogeneity between ENSO teleconnections and dengue epidemics
The number of dengue cases exhibited a predominantly increasing trend in endemic countries across the Americas and Asia between 1980 and 2019. Throughout this period of escalation, dengue outbreaks frequently occurred in El Niño years, with a particularly significant correlation with ENSO fluctuations in the recent decade ($r = 0.68$, $P < 0.05$) (Fig. 1a and Supplementary Fig. 1a). Although long-term correlations between the time series of ENSO indices and annual dengue cases are insignificant in most countries, there exists a great magnitude of dengue epidemics in endemic countries such as Brazil,

Vietnam, and Indonesia strongly affected by El Niño, i.e., more cases during El Niño years but fewer cases during La Niña years, as evidenced by both the linear trend and the clustering of endemic countries above the one-to-one line in Fig. 1b. However, the association between the time series of ENSO indices and annual dengue cases varies greatly across countries (Supplementary Fig. 1b).

To disentangle the heterogeneity in the association between ENSO and dengue epidemics, we investigated the teleconnections between ENSO indices and population-weighted climate averages across countries (Fig. 1c). ENSO teleconnections, measured as partial correlations of ENSO with local temperature and precipitation, allow us to isolate the impact of ENSO-driven temperature and precipitation from seasonal averages. ENSO displayed opposite teleconnections with local temperature and precipitation in most dengue-endemic countries (39/57, Fig. 1d and Supplementary Fig. 2c). Specifically, ENSO exhibited positive teleconnections with local temperature but negative teleconnections with local precipitation in 34 countries, including Vietnam and Thailand. Inversely, ENSO displayed negative teleconnections with temperature and positive teleconnections with precipitation in five countries including Mexico. Aligned teleconnections, in which both ENSO-temperature and ENSO-precipitation teleconnections are positive, were found in the remaining 18 countries, including Brazil and Singapore. These discrepancies in teleconnections highlight the complex association between ENSO (hereafter represented by the DJF NINO3.4 index) and global dengue dynamics.

### Persistent effects of ENSO teleconnections on dengue epidemics
To explore the impact of ENSO on dengue epidemics, we built a country-level distributed lag regression model using the aforementioned ENSO teleconnections (Methods). Our results illustrated that ENSO could explain 63% of the variation in interannual dengue cases after controlling for the country-level heterogeneity (Supplementary Table 1). The intensity of ENSO's effects on dengue epidemics is associated with the strength of teleconnections (Fig. 2a, b). ENSO primarily increased dengue risk (represented by annual dengue cases) via rising local temperature. Conversely, its impact via local precipitation was smaller, and it varied in direction. A 1 °C increase in the NINO3.4 index through positive ENSO–temperature teleconnections was associated with an averagely 48.0% increase in annual dengue cases (95% CI = 9.5%–140.5%) in the occurrence year. Simultaneously, a 1 °C increase in the NINO3.4 index through either positive or negative ENSO–precipitation teleconnections was associated with an averagely 12.2% increase (95% CI = 1.0%–27.2%) or a 13.1% decrease (95% CI = 1.3%–26.0%) in annual dengue cases, respectively (Fig. 2c).

To validate whether ENSO persistently affects dengue epidemics, we estimated the magnitude of lagged effects of ENSO teleconnections on dengue epidemics (Fig. 2a, b) and found consistent effects up to 2 years after the event (Supplementary Fig. 3, owing to herd immunity, time lag up to 2 years was considered), suggesting the impact of ENSO is not constrained to the season in which it first occurs. We also examined non-linear functions between ENSO teleconnections and dengue epidemics and revealed that most observations followed a linear relationship, with only a few extreme values deviating from this pattern (Supplementary Fig. 4 and Supplementary Table 2). This suggests that ENSO-driven temperature is currently below the thermal optima for mosquito activity in most areas[35,42]. Sensitivity analysis indicated that the effect of ENSO via temperature was relatively robust to seasons, dominant mosquito species, and economic status compared with its effect via precipitation (Supplementary Figs. 5–7 and Supplementary Table 1). Additionally, the effects remained relatively stable after incorporating data from temperate countries or accounting for potential time-varying confounders, such as changes in WHO case definitions in 2009 and variations in circulating serotypes (Supplementary Fig. 8 and Supplementary Table 3). The aforementioned analysis suggests that ENSO teleconnections are persistently

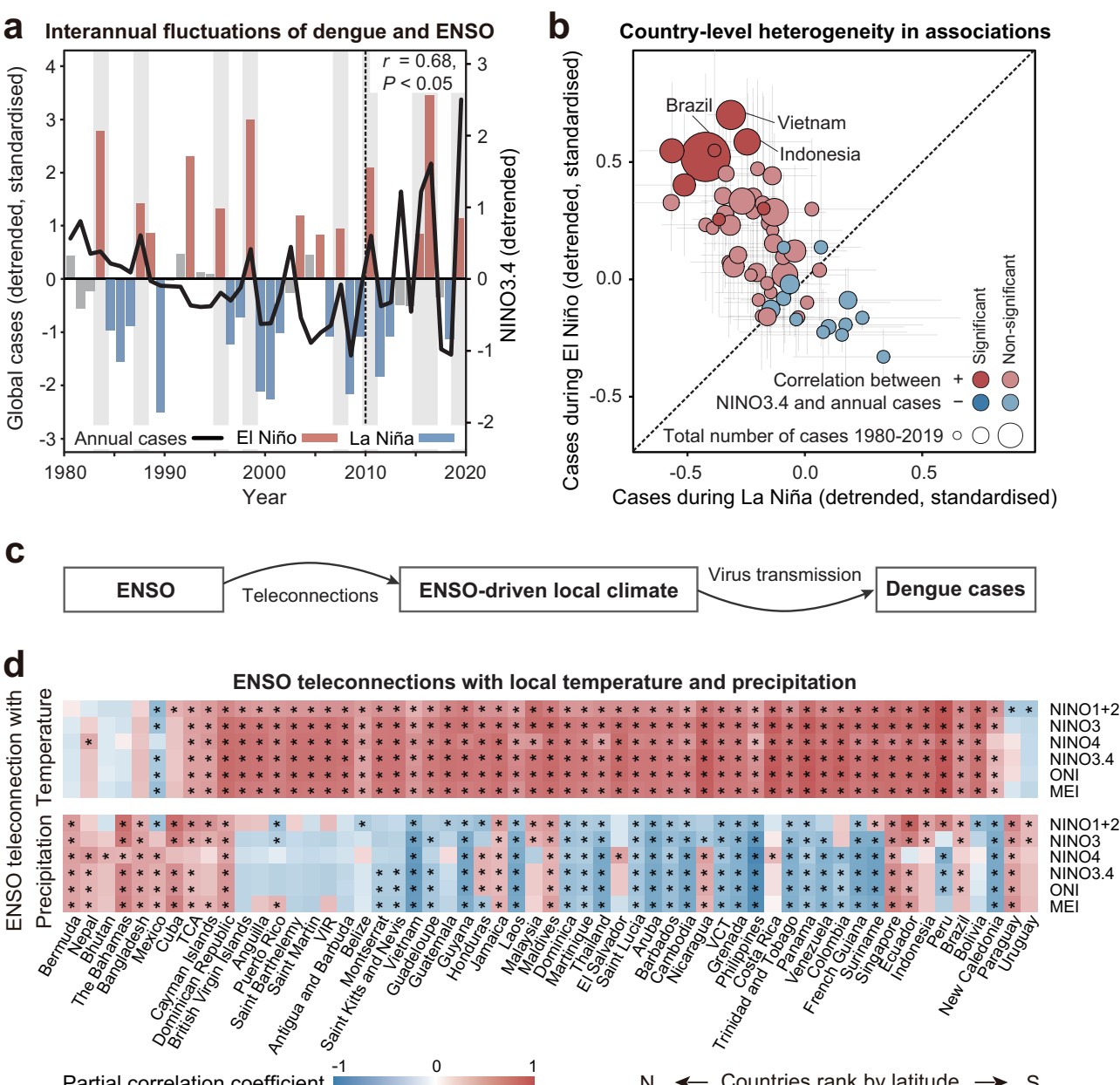

**Fig. 1 | Association between ENSO and dengue epidemics in 1980–2019.**
**a** Interannual time series of global dengue cases (black line) and the DJF NINO3.4 index (calculated as the December–February average of sea surface temperature anomalies over the region at 5°N–5°S and 170°W–120°W; bars). El Niño events (red bars) and La Niña events (blue bars) are represented by the DJF NINO3.4 index being above and below the thresholds of +0.5 °C and −0.5 °C, respectively, and the remaining were neutral events (grey bars)[66]. The grey shading indicates dengue outbreaks occurred in El Niño years. The dashed line highlights the recent decade with the strongest correlation coefficient ($r = 0.68$) and its significant level based on a two-tailed test ($P = 0.03$) between detrended DJF NINO3.4 and global dengue cases compared to previous decades. Source data are provided as a Source Data file. **b** Comparison of annual dengue cases during El Niño and La Niña years in each country. Dots represent the average annual number of cases for a country ($n = 57$), and error bars denote the standard deviation. The dot colour denotes the Pearson correlation coefficients between ENSO (represented by the DJF NINO3.4 index) and annual dengue cases: red for positive, blue for negative, with darker colours indicating significant correlations ($P \leq 0.05$). +: positive; −: negative. The dot size represents the total number of standardised dengue cases reported from 1980 to 2019. Source data are provided as a Source Data file. **c** Schematic of the proposed mechanism of ENSO's effects on dengue cases through its teleconnections with local climates. **d** Teleconnections measured as the maximum partial correlation between DJF ENSO indices and local temperature (upper panel) and precipitation (lower panel) across 3-month running intervals from December to May in each country (Methods). All variables were detrended and standardised. The correlation coefficient ranges from negative (blue) to positive (red). Significant correlations based on two-tailed tests ($r > 0.304, P \leq 0.05$) are denoted by an asterisk. TCA Turks and Caicos Islands, VIR United States Virgin Islands, VCT Saint Vincent and the Grenadines. Source data are provided as a Source Data file.

associated with dengue epidemics in two types via temperature and precipitation: trade-off (opposite effects via temperature and precipitation) and synergy (aligned effects via temperature and precipitation; Fig. 2d). Most countries (39/57) experienced trade-off in ENSO's impacts on dengue cases.

By quantifying the cumulative effects of ENSO on dengue risk for each country, we observed that countries experiencing opposite effects of ENSO teleconnections via temperature and precipitation (trade-off effects) were mainly distributed in Southeast Asia and western South America, and aligned effects (synergy effects) occurred in

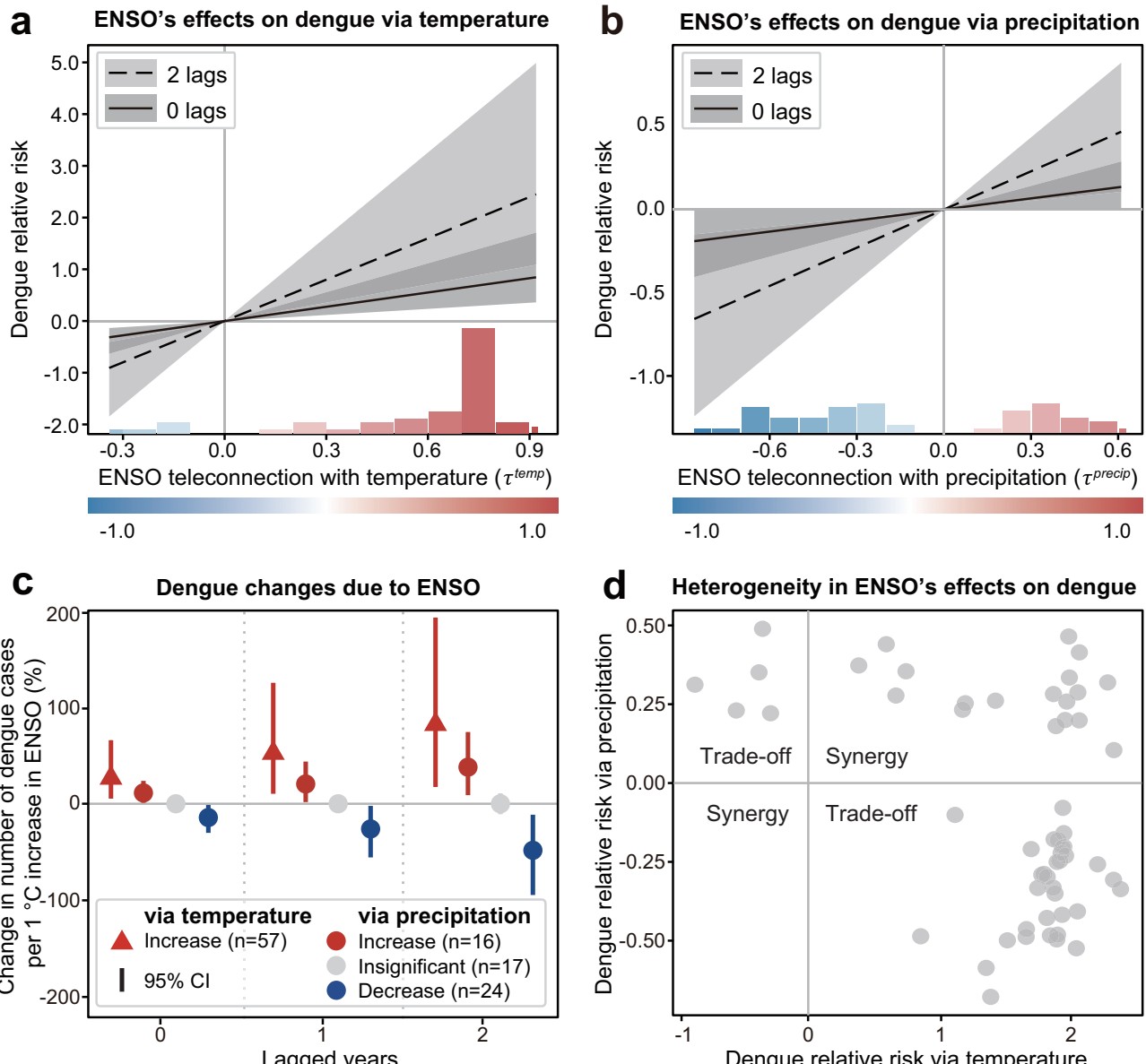

**Fig. 2 | Persistent effects of ENSO on dengue epidemics through teleconnections.** Effects of ENSO teleconnections on dengue cases via temperature (**a**) and precipitation (**b**). The solid line represents the effects in the occurrence year of the event (zero lags), and the dashed line represents the cumulative effects from the year of the event to the second year after the event (two lags). Lines denote mean values, and shaded areas denote the 95% CIs from 1000 bootstrap re-samplings. The histogram along the bottom presents the distribution of country-level teleconnections, with the colour indicating the direction and strength of the teleconnection. Source data are provided as a Source Data file. **c** Cumulative changes in the number of dengue cases per one-unit increase in the NINO3.4 index via temperature (triangle) and precipitation (circle) in the occurrence year, the following year, and the second year. The change via precipitation was drawn by country groups based on the direction of ENSO–precipitation teleconnections, and the change via temperature was drawn together because most countries exhibited positive values, with "n" denoting the number of countries. Dots denote mean values, and lines indicate 95% CIs. Source data are provided as a Source Data file. **d** Trade-off and synergy effects of ENSO teleconnections on dengue epidemics. Trade-off represents opposite effects of ENSO teleconnections via temperature and precipitation, and synergy represents aligned effects. Source data are provided as a Source Data file.

Indonesia, Southern Asia, and eastern South America (Fig. 3a, b, left panel). Furthermore, countries in the Caribbean, Central America, South America, and Southeast Asia experience strong ENSO effects primarily via temperature, whereas countries in Northern America and Melanesia experience these effects via precipitation (Fig. 3b, left panel). Additionally, countries with higher local temperatures can simultaneously encounter stronger positive teleconnections via temperature and stronger negative teleconnections via precipitation (Fig. 3b, middle panel). These regions, in which dengue was strongly affected by ENSO, have been widely acknowledged as endemic areas

and regions with highly favourable environmental conditions for *Aedes* mosquito vectors[32].

We further analysed the impact of the four strongest El Niño events on dengue epidemics over the past 45 years (Methods). The El Niño events of 1982–83, 1997–98, 2015–16 were predicted to induce an additional 0.2 (95% CI = 0.1–0.3), 1.4 (95% CI = 0.9–1.8), and 4.1 million dengue cases (95% CI = 2.8–4.8) within 2 years after the event, respectively (Supplementary Fig. 9a, b). Since the effects are robust when including the data for 2023 and 2024 (Supplementary Fig. 8c, d), the recent 2023–24 El Niño event was predicted to induce an

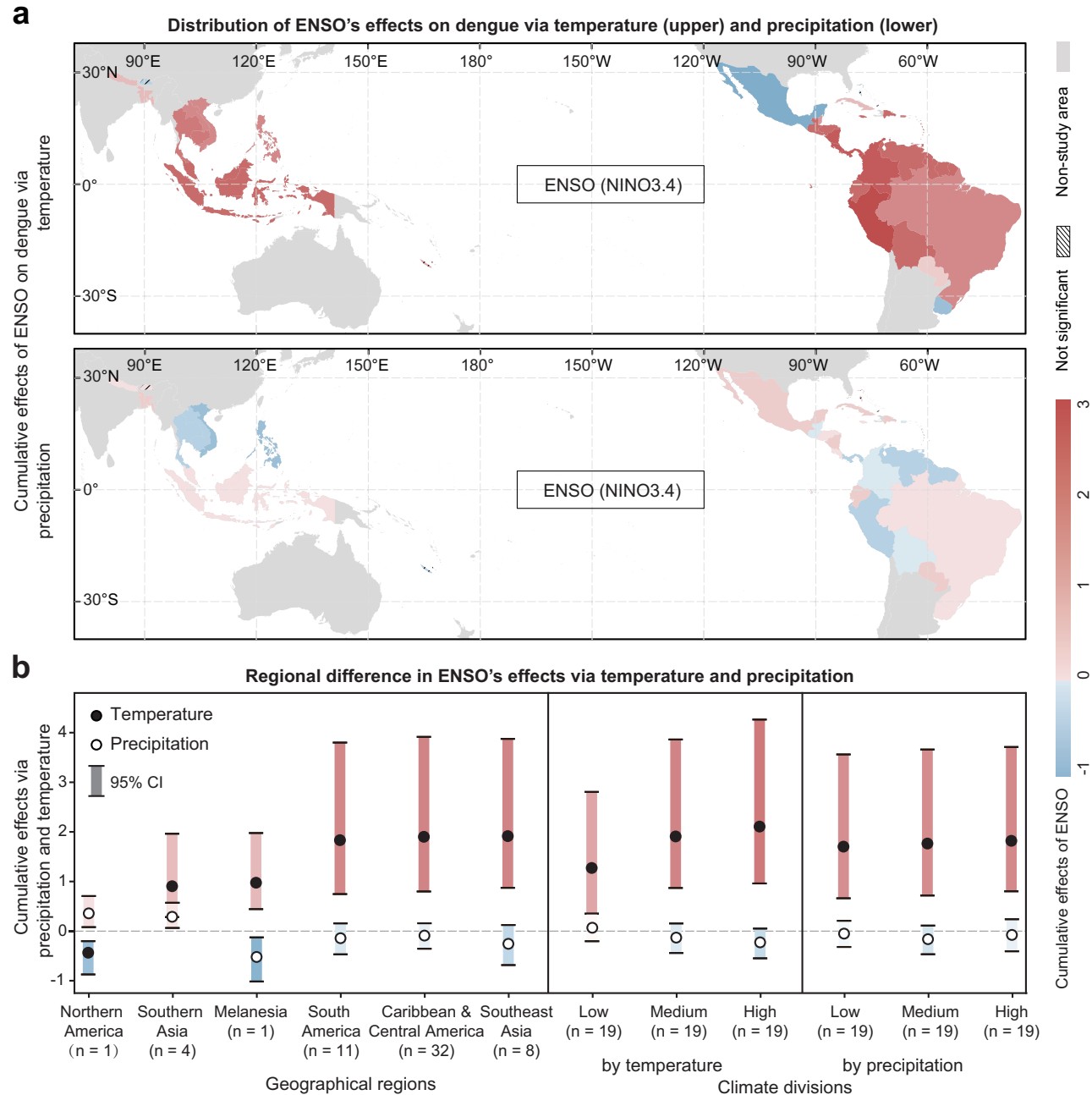

**Fig. 3 | Regional heterogeneity in the effects of ENSO on dengue risk.**
**a** Distribution of the cumulative effects of ENSO on dengue epidemics via temperature (upper) and precipitation (lower). Combining the two maps, opposite effects via temperature and precipitation (red and blue, respectively) indicate a trade-off, and aligned effects (both red) indicate synergy. The effects of ENSO are not significant in Bhutan, New Caledonia, Bahamas, and Bermuda. Source data are provided as a Source Data file. **b** Regional differences in ENSO's effects on dengue risk via temperature and precipitation. Countries are grouped by geographical regions (left panel) and climate conditions according to one-third quantiles of annual mean temperature (middle panel; low temperature: 14.6–24.9 °C, middle temperature: 25.1–26.1 °C, high temperature: 26.1–27.9 °C) and annual cumulative precipitation (right panel; low precipitation: 665.5–1095.5 mm, middle precipitation: 1133.9–1708.9 mm, high precipitation: 1767.6–3650.1 mm) for study countries, with "n" denoting the number of countries in each group. Dots indicate average values, and bars denote 95% CIs. Source data are provided as a Source Data file.

additional 9.6 million cases (95% CI = 6.3–12.6) in the occurrence year, with potentially more cases in 2025 and 2026. The relatively greater effects of the 2015–16 and 2023–24 events were probably attributable to a stronger El Niño, an increasing trend in the number of global dengue cases, and the concentration of cases in countries experiencing synergy effects of ENSO via temperature and precipitation. Because an El Niño event always triggers a subsequent La Niña event, we included the negative effects of the following La Niña and found that the impacts of these El Niño events have been smaller. For

instance, the 2015–16 El Niño induced 3.2 million cases (95% CI = 2.1–3.6) when accounting for the 2018 La Niña (Supplementary Fig. 9b).

## Effects of future ENSO changes on dengue epidemics
The complex and region-specific influence of ENSO on dengue risk raises questions about how changes in ENSO will shape future dengue risk in a warmer world. We first projected changes in ENSO and teleconnections using data from the sixth phase of the Coupled Model

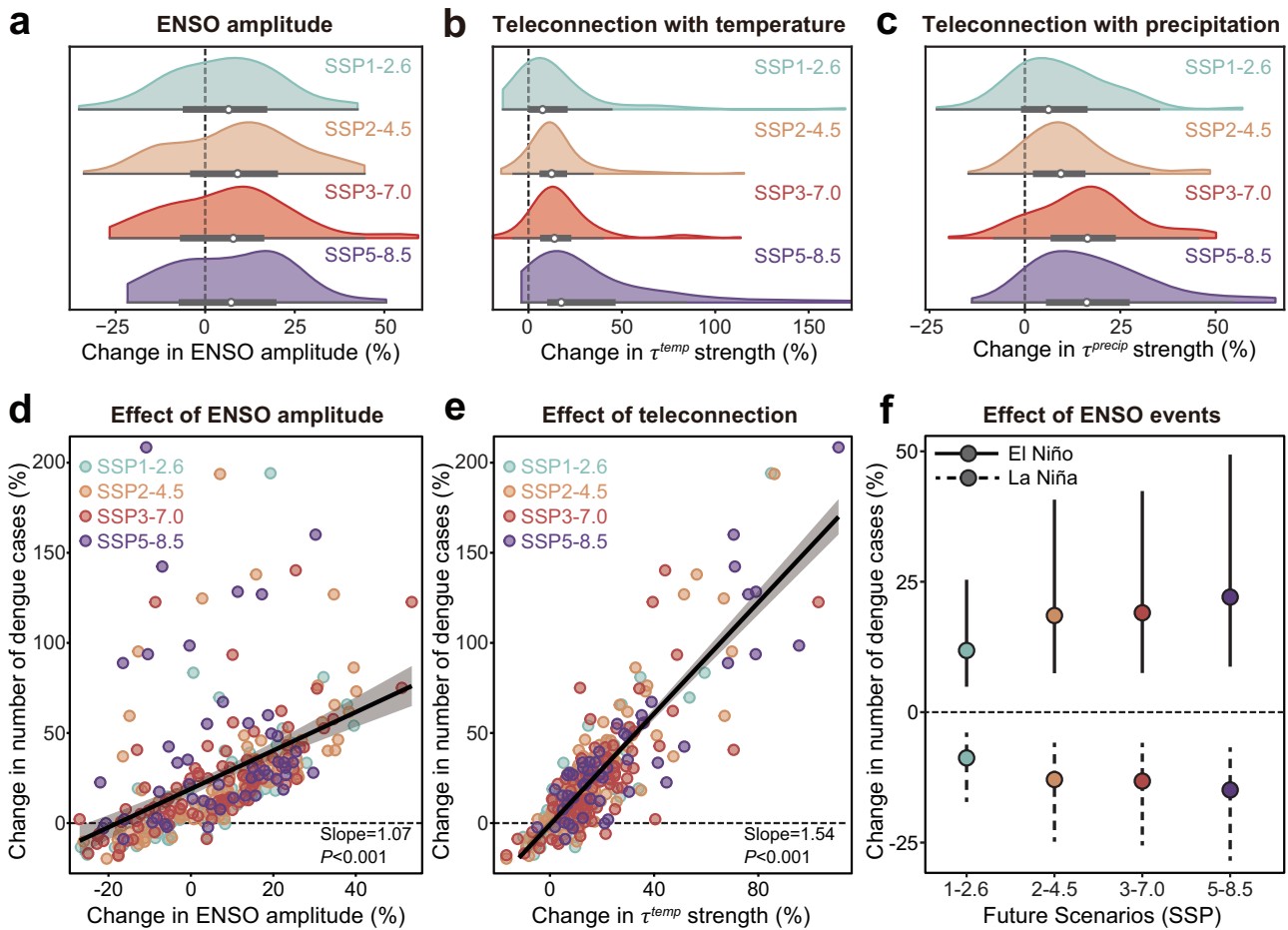

**Fig. 4 | Future ENSO change and its effects on the global dengue burden.**
Changes in the ENSO amplitude (**a**) and global mean teleconnection strength (the absolute value of teleconnection) with local temperature (**b**) and local precipitation (**c**) between 1940–2019 and 2020–2099 for multiple ensemble members of CMIP6 simulations from four scenarios. The ENSO amplitude was measured as the standard deviation of the DJF NINO3.4 series. We selected 69 out of 88 ensemble members for SSP1-2.6, 105 out of 150 for SSP2-4.5, 101 out of 134 for SSP3-7.0, and 58 out of 94 for SSP5-8.5 because the bias between simulated amplitude and observed amplitude in 1980–2019 is less than 50% of the observed amplitude[77]. Error bars indicate the median and the 25th–75th percentiles. Source data are provided as a Source Data file. Changes in global dengue cases due to changes in ENSO amplitude (**d**) and teleconnection strength with temperature (**e**). Changes in global dengue cases were calculated as the change rate in dengue cases between projected future (2020–2099) and counterfactual future without ENSO evolution under human-induced warming (Methods). Each dot corresponds to one CMIP6 simulation. Lines denote mean values, shaded areas denote the 95% CIs, and the regression lines and 95% CIs are drawn for all simulations and scenarios. The significant level is assessed by a two-tailed test. Source data are provided as a Source Data file. **f** Changes in global dengue cases due to changes in El Niño and La Niña events between projected future and counterfactual future. The calculation was based on the median values of projected changes in the ENSO amplitude and teleconnection strength across CMIP6 simulations. Dots denote averages, and bars indicate the 95% CIs from 1000 bootstrapping of regression coefficients. SSP1-2.6, low-emission scenario, the Paris Agreement to limit warming to 1.5 °C–2.0 °C above the pre-industrial level; SSP2-4.5, middle of the road, following historical trends; SSP3-7.0, a fragmented world with regional differences in climate policy; and SSP5-8.5, high-emission scenario, no additional climate policy[39]. Source data are provided as a Source Data file.

Intercomparison Project (CMIP6), considering 333 ensemble members of 39 climate models under four shared socioeconomic pathways (SSPs; Supplementary Table 4). Our projections illustrated that, to a certain extent, both the amplitude and teleconnections of ENSO may increase from 1940–2019 to 2020–2099 under the warming context (Fig. 4a–c), consistent with previous research[43–47]. The median amplitude was projected to rise by 6.6–9.0% across scenarios, with global median teleconnections increasing by 7.9–18.1% for temperature and 6.3–17.2% for precipitation. However, these predictions are shaped by stochastic projections of ENSO and related climate variability–34.5% of ensemble members predicted a decrease in ENSO amplitude and future ENSO patterns could be dominated by either El Niño or La Niña[31].

We combined these projections with our empirical estimates to quantify the future (2020–2099) changes in dengue risk. We defined the baseline as a "counterfactual future" in which ENSO remains within the observed amplitude and teleconnections in 1980–2019 without the influence of human-induced warming. Despite the range of projections being large across models and scenarios (Supplementary Fig. 10), we projected that future changes in ENSO would be systematically related to increased dengue risk in the future (Fig. 4d, e). Each additional 1% increase in the ENSO amplitude is expected to be associated with a 1.07% increase in dengue risk ($P < 0.001$, Fig. 4d), and each additional 1% increase in the ENSO teleconnection with temperature is expected to be associated with a 1.54% increase in dengue risk ($P < 0.001$, Fig. 4e). Some endemic countries, such as Peru and Malaysia, could experience greater than 90% increases in dengue cases because of ENSO under SSP5-8.5 in the future (Supplementary Fig. 11).

More specifically, we estimated that intensified El Niño events in the future would induce more cases, whereas intensified La Niña could

slightly mitigate the number of cases (Fig. 4f). Intensified El Niño events could lead to 11.8–22.1% increases in global dengue cases in the occurrence year and 39.0–81.7% increases in cumulative cases when accounting for lagged effects (two lags), indicating a range from low- to high-emission scenarios (Fig. 4f). Owing to the non-linear response of dengue transmission to temperature[35,48], there is a risk that we may over-estimate dengue risk when local temperatures exceed the thermal optima under warming. To address this, we estimated average summertime temperature in the 21st century and found that it is unlikely to exceed the thermal optimum in 98.3% ($n = 56$), 91.2% ($n = 52$), 89.5% ($n = 51$), and 77.2% ($n = 44$) of endemic countries under SSP1-2.6, SSP2-4.5, SSP3-7.0, and SSP5-8.5 scenarios, respectively (Supplementary Fig. 12). Additionally, mosquitoes possess wide thermal safety margins through their use of shaded microhabitats[49], indicating that the linear relationship between temperature and dengue risk could persist in the future. These findings imply that future ENSO will likely increase the amplitude of global dengue epidemics.

## Discussion

Building upon our previous study demonstrating the ability of global climate indices to predict dengue incidence[50], this study provides quantification of the effects of global climate variability, represented by ENSO, on future dengue epidemics. By decomposing the region-specific effects of ENSO on local temperature and precipitation anomalies based on teleconnection metrics and distributed lag model, we are able to better understand the link between ENSO events and dengue risk. We further show that projected climate changes are likely to increase the impact of ENSO on dengue with larger epidemics, a new risk that must be mitigated against. While previous research has observed occasional associations between ENSO and dengue risk, this study attempts to assess the relationship at a global scale using long-term disease and climate data, and the first to reveal the mechanism of this association. As ENSO teleconnections we assessed are independent of long-term climate change, our understanding based on teleconnections can be used to develop medium- and long-range outbreak forecasting systems to prevent and mitigate dengue outbreaks, and to empower studies that attribute the effects of climate change on current epidemics.

Our study demonstrates that ENSO regulates dengue risk mainly through rising local temperatures. This mechanism aligns with extensive climate-dengue research and laboratory-based evidence showing the relationship between local temperature and dengue risk due to its impact on mosquito survival, biting, reproduction and viral replication rates[35,37]. In contrast to the role of temperature in enhancing dengue transmission when it is below the thermal optima for mosquitoes, the effect of precipitation on dengue transmission is more intricate. Drought reduces breeding habitats which lowers mosquito density, but it also increases artificial habitat from stored water and mosquito migration to urban areas with more potential hosts[37,51]. Although our linear approaches are limited in capturing how monthly rainfall variation affects dengue transmission[52,53], our study focuses on assessing the effects of ENSO fluctuations on annual dengue cases through local temperature and precipitation. By integrating ENSO teleconnection patterns, our analysis reveals that the significant impact of ENSO likely occurs via direct heat transfer from the ocean, rather than through affecting local precipitation. Notably, this significant impact pathway does not conflict with the nonstationary relationship between the time series of ENSO and dengue cases which might be explained by the transient nature of coupled climate-disease interactions and interference from compounding drivers[3,16]. Our finding implies that bridging climatic processes with laboratory-based observations on vector temperature and rainfall tolerance would further improve our understanding of the mechanisms by which global climate change shapes dengue epidemics.

Our analysis confirms the consistency of the association between ENSO and elevated dengue risk at the global level, despite regional variations in magnitude. Previous studies observed that dengue cases peak 3–6 months following an El Niño event[20,54]. Our findings further suggest that this effect can persist for up to 2 years. This prolonged effect of ENSO might be attributed to epidemic momentum, where substantial transmission persists through the low season in areas where it would normally not, or longer-term adaptive changes in mosquito feeding and egg-laying behaviour and population dynamics after ENSO-induced local climate changes[55,56]. The persistent effect highlights the necessity for public health and emergency systems to consistently monitor and respond to ENSO impacts. Our findings could provide valuable information for mitigating the effects of climate change on dengue epidemics. Further research on the long-term adaptations of mosquito populations triggered by ENSO is needed to validate this persistent effect.

Our findings underscore the region-specific role of ENSO in driving dengue epidemics. We found that ENSO has a particularly strong impact on dengue cases in endemic areas and regions with favourable environmental conditions for *Aedes* mosquito vectors, which is possibly attributed to the strong teleconnections between ENSO and local climates. This contributes to existing climate–dengue research[41,57] by reinforcing the necessity of maintaining robust preparedness strategies for ENSO events in traditional endemic areas. Furthermore, our analysis suggests that ENSO similarly influences dengue cases in non-endemic areas, including temperate countries. As the mosquito distribution expands with global warming[58–60], our findings emphasise the need for temperate countries to be vigilant about potential dengue outbreaks during El Niño events.

Future prospects highlight the unrecognised impact of global climate variability on the magnitude of dengue epidemics, which is independent of long-term warming trends. Although future ENSO projections remain debatable, our predictions based on 333 multi-model ensemble members indicate an increasing risk of dengue transmission driven by intensifying ENSO events and their teleconnections under global warming. Our estimates differ from existing projections based on local climate and socioeconomic predictors[59] because controlling for these variables in our regression analysis did not alter the effect of ENSO. The consistent increase in global dengue risk across all emission scenarios implies that long-term emission reductions alone could be insufficient to mitigate dengue outbreaks driven by ENSO. Enhanced short-term interventions, such as disaster risk management informed by ENSO early warnings, might offer more effective control measures to mitigate the severity of future outbreaks.

Several limitations relevant to the findings exist in our data source and methodology. First, our model did not incorporate detailed mechanisms by which temperature and precipitation affect transmission, potentially underestimating the importance of ENSO-driven precipitation on mosquito breeding. Second, although we examined the effects of ENSO-driven temperature and precipitation on dengue cases in isolation, potential confounding effects, such as increased vector control activities or human behaviour changes that affect exposure to human biting, were not considered which might underestimate the comprehensive effect of ENSO on dengue cases. Third, our country-level teleconnections may overlook subnational heterogeneity in ENSO-dengue relationships particularly for large countries. Future studies with access to higher-resolution case data could build upon our findings by examining these subnational patterns such as city-level associations. Forth, our exclusion of Africa due to surveillance disparities may impact the understanding of global ENSO-dengue mechanisms. Given Africa's substantial dengue burden, developing surveillance networks is needed to establish complete global mechanisms. Fifth, uncertainties in our results may also stem from the potential of exceeding the thermal optima for mosquitoes

under future warming scenarios, as well as variations in measurement methods and definitions of reported dengue cases across countries. Additionally, while we showed insignificant correlations between switches in dominant dengue virus (DENV) serotype and annual cases in this analysis, a lack of global serotype data meant we were unable to disentangle the virus and climate-attributable causes of large epidemics. Future research is recommended to account for other factors such as economic development, land use changes, serological status of the population, and viral evolution to better explain interannual dengue variations.

Overall, our study highlights the importance of integrating ENSO-driven local climate factors to develop an understanding of the relationship between ENSO events and dengue epidemics. These findings have revealed the unrecognised impact of global climate variability and could empower specific countries to better prepare for and effectively respond to potential dengue outbreaks during future ENSO events.

## Methods

### Data

**Dengue case data**. Annual reported dengue cases used in this study were compiled from 57 countries covering the period from 1980 to 2024 (Supplementary Fig. 13 and Supplementary Table 5). We mainly assembled this global dengue dataset from the OpenDengue website (https://opendengue.org/)[61], with a time resolution set to "Year". We also conducted an extensive search and cross-verified data across various sources, including the World Health Organization (WHO), the Pan-American Health Organization (PAHO), online databases, authorised Ministry of Health websites, established datasets[50], and published literature. For countries with missing data such as Thailand, Malaysia, Singapore, New Caledonia, Bhutan, China, and Maldives, we accessed their Ministry of Health or equivalent websites to fill in the gaps. Data from these sources were aggregated from weekly, monthly, and quarterly intervals to construct yearly datasets. Annual case counts for 2023 and 2024 were estimated based on the most complete annual cases for 2023 and 2024 as of the reporting date of 1st March 2025. Due to reporting delays, Anguilla, Bhutan, Dominica, and Venezuela were yet to finalise their case counts for 2024 by this date. We thus used a simple proportion of annual total cases by month model to predict their final 2024 annual case counts (Supplementary Text).

Note that data from 2020 to 2024 were not included in the main model training to avoid the impact of COVID-19. We included the recent years 2023 and 2024 in the sensitivity analysis to demonstrate the robustness of our main model. We also collected data from four temperate countries—Argentina, China, Chile, and the United States in our sensitivity analysis. Despite inter-country variations in case definitions and surveillance quality, intra-country variations over time can still be used to understand the drivers of dengue transmission.

### ENSO indices, local climate, and population data

To assess the El Niño–Southern Oscillation (ENSO) events, we acquired the 1978–2024 time series of six ENSO indices (Supplementary Fig. 1a) from the National Climate Center (NCC) of China and the National Oceanic and Atmospheric Administration (NOAA) of the United States. Local climate observations were obtained from ERA5 monthly mean reanalysis product with a resolution of $0.25° \times 0.25°$, including air temperature at 2 m above the Earth's surface and precipitation from 1980 to 2024[62]. The Gridded Population of the World (GPWv4) data at the 15 arc-minute resolution[63] was utilised to calculate the population-weighted climate for each country. We also collected country-level population statistics for the 1980–2024 time series from the World Population Prospects[64] for the offset term in our regression model.

### Population-weighted climate calculation

We used population-weighted local climate to accurately capture the climate variations that affect people and dengue transmission. Population weighting gives more weight to areas with higher population density, ensuring that the climate conditions experienced by the majority of the population are better represented[31,65]. This approach enhances the statistically valid assessment of climatic effects on dengue cases, allowing for more targeted public health interventions and improved resource allocation based on where people are most affected. Thus, based on gridded population data, we derived population-weighted averages of climatic variables:

$$PrEnv_i = \frac{\sum_{g=1}^{n} N_g \times Env_g}{\sum_{g=1}^{n} N_g} \tag{1}$$

where $PrEnv_i$ represents the population-weighted climate of country $i$, $N_g$ and $Env_g$ denote the population and climate variables of across grid $g$, respectively, and $n$ denotes the total number of grid cells in country $i$.

### Correlation analysis

We calculated the Pearson correlation coefficient between ENSO indices and annual dengue cases. To ensure that the time series of ENSO indices precede the occurrence of the disease, we scrutinised the temporal correspondence for the north and south hemispheres. We conducted a linear detrend algorithm and z-score standardisation for ENSO indices and dengue cases to remove the effects of long-term warming trends.

We further assessed the collinearity between the annual changes in six ENSO indices based on the Pearson correlation coefficient and identified the NINO3.4 index as the most representative ENSO indicator. NINO3.4 is defined as the averaged anomaly of sea surface temperatures (SSTA) in the tropical Pacific Ocean over 5°S–5°N and 170°W–120°W. El Niño and La Niña events are characterised by a five-consecutive three-month running mean of the Niño 3.4 index exceeding +0.5 °C or falling below −0.5 °C, respectively[66,67]. To focus on the season of peak ENSO activity, we used the average NINO3.4 over winter months for the following analysis between ENSO and dengue cases. Specifically, ENSO in a given year $t$ was represented by the average NINO3.4 values for December of the preceding year $t$-1 and January and February of the current year $t$ (referred to as DJF NINO3.4).

### ENSO teleconnections with country-level climate

To match the country-level disease data resolution, we measured ENSO teleconnections as the extent to which each country's climate is influenced by ENSO, accounting for the effects of temperature and precipitation, and different time scales at which teleconnections may manifest. Unlike previous approaches that examine the impacts of global climate and local climate on dengue epidemics separately, this method allows for a comprehensive study of the impact of ENSO-driven local climate on dengue epidemics, providing a more nuanced understanding of their relationships. Although ENSO may have opposing effects within large countries like Brazil[68], this study focuses on assessing the long-term effects of ENSO on country-level dengue cases due to data availability.

We used partial correlation to evaluate the teleconnections between interannual changes in DJF NINO3.4 and the local climatic variables for each country. First, we processed the 3-month running values of local climatic variables to eliminate random variations and ensure consideration of ENSO exposure over multiple months, including December$_{t-1}$–January$_t$–February$_t$ (DJF), January$_t$–Ferurary$_t$–March$_t$ (JFM), Ferurary$_t$–March$_t$–April$_t$ (FMA), and March$_t$–April$_t$–May$_t$ (MAM).

For each subset series of 3-month running mean temperature and cumulative precipitation during 1980–2019, we conducted a linear detrend algorithm and z-score standardisation to remove the effects of long-term warming trends and rare climatic events. Next, we calculated the partial correlation coefficients between these detrended and standardised time series of local climatic variables and DJF NINO3.4. Partial correlation was employed when analysing the ENSO–temperature correlation to control for precipitation, and vice versa, to account for the interactions between temperature and precipitation. This calculation yielded 4 correlation coefficients (DJF, JFM, FMA, MAM) for each country, separately for temperature and precipitation. We then identified the maximum absolute value of the correlation coefficients for each country. Finally, the positive and negative signs were reassigned to the maximum absolute values correspondingly to obtain the ENSO teleconnections with local temperature ($\tau^{temp}$) and precipitation ($\tau^{precip}$). The obtained ENSO teleconnections were then integrated into the ENSO-climate-dengue model. To assess the sensitivity of our findings to our chosen period over which ENSO teleconnections were calculated, we also calculated alternative teleconnection metrics over the summer, the whole year, and all the 3-month running values between December ($t$-1) and May ($t$). To address multiple hypotheses and control the false discovery rate, we employed the Benjamini-Hochberg (BH) method to assess the significance of the correlation coefficients[69].

### ENSO–climate–dengue model
Our analysis aims to quantify the effects of ENSO on dengue epidemics, which requires us to distinguish ENSO from other confounding factors. We employed a distributed lag regression model using ordinary least squares estimation to estimate the effects of ENSO on dengue cases:

$$\log(\text{case}_{it}) = \sum_{L=0}^{j} \left( \beta_L \text{NINO3.4}_{t-L} + \left( \theta_L^{temp} \text{deNINO3.4}_{t-L} * \tau_i^{temp} \right. \right.$$
$$\left. \left. + \theta_L^{precip} \text{deNINO3.4}_{t-L} * \tau_i^{precip} \right) \right) \quad (2)$$
$$+ \text{offset}(\text{population}_{it}) + \mu_i + \epsilon_{it}$$

In Eq. 2, $\text{case}_{it}$ denotes the reported cases in country $i$ in year $t$. $\text{NINO3.4}_t$ represents the NINO3.4 index in year $t$ with the linear trend, while $\text{deNINO3.4}_t$ represents the detrended NINO3.4 index in year $t$. Given that ENSO primarily drives local climate anomalies and extremes through oscillations[25,70], the detrended NINO3.4 index was used to unravel the effects of ENSO on dengue via local climate. This approach enables us to capture the oscillatory nature of ENSO while avoiding the repetition of multiple signals related to global warming in the equation. $\tau^{temp}$ and $\tau^{precip}$ allow us to isolate the impact of ENSO-driven temperature and precipitation on dengue epidemics from each other and seasonal averages. $L$ denotes the lagged year in which the coefficient is estimated. Here, we estimated the impact of ENSO events in the current year, subsequent year and the year after that (i.e. a 2-year lag, $j = 2$) allowing us to capture longer-term indirect impacts of ENSO that may be triggered by ongoing epidemic momentum or longer-term environmental adaptations by the human and vector populations in response to the ENSO-induced extreme climate conditions[56,71,72]. We tested the multicollinearity and confirmed low correlations among multiple lags and ENSO-driven climatic factors (Supplementary Table 6). $\text{population}_{it}$ is the total population in country $i$ in year $t$, serving as an offset term. $\mu_i$ is a time-invariant country fixed effect, which controls for differences among countries' ability to control, detect and report dengue cases. We did not include year fixed effects in our main analysis due to their high collinearity with the NINO3.4 index, but we tested the influence of important year-varying confounders that may affect dengue cases (i.e., the changes in WHO case definition in 2009 and yearly variations in the circulating serotype in 9 countries from 1990 to 2019

based on the dengue sequences reported in the Bacterial and Viral Bioinformatics Resource Center[73]).

Our analysis focused on the cumulative coefficients $\Omega_{iL}^{temp}$ and $\Omega_{iL}^{precip}$, representing the accumulated effects of ENSO on reported dengue cases for each country $i$ in $L$ years after the event, via temperature and precipitation, respectively:

$$\Omega_{iL}^{temp} = \sum_{L=0}^{j} \left( \theta_L^{temp} * \tau_i^{temp} \right) \quad (3)$$

$$\Omega_{iL}^{precip} = \sum_{L=0}^{j} \left( \theta_L^{precip} * \tau_i^{precip} \right) \quad (4)$$

These influences differ between countries and lag lengths, depending on the strength of the coupling between each country's temperature and precipitation with ENSO. The inclusion of lags from year $L$ to year $j$ helps differentiate between transient and persistent impacts of ENSO on dengue cases. If the coefficient is significantly different from zero ($P \leq 0.05$), it suggests that ENSO has a persistent impact on dengue cases. Conversely, if it is not significantly different from zero, we cannot reject the hypothesis and conclude that ENSO only affects transient dengue outbreaks. Confidence intervals were estimated using 1000 bootstrap samples. We also estimated the confidence intervals using a Bayesian generalised linear model with 2000 bootstrap samples, and obtained similar results (Supplementary Table 7).

Despite the inclusion of lagged terms, the Ljung-Box test ($Q^* = 2465.8$, $P \leq 0.01$) and the Breusch-Pagan test ($BP = 46.3$, $P \leq 0.01$) indicated that temporal autocorrelation and heteroscedasticity still exist in our model. To avoid the impact of temporal autocorrelation, we applied the Newey-West adjustment[74] to recalculate standard errors of the estimates. The Newey-West adjustment has been commonly used in econometric analysis and epidemiological studies[75] to ensure the validity of statistical inference. The adjusted standard errors validated that temporal autocorrelation does not affect the significance of our regression coefficients (Supplementary Table 1). We also included $\text{case}_{t-1}$ as an explanatory variable to control the autoregressive effect and validated that our regression results remain robust.

### Magnitude of dengue epidemics from historical extreme El Niño events
The regression coefficients derived from Eq. 2, $\theta_L^{temp}$ and $\theta_L^{precip}$, not only provide estimates of the change in dengue cases per one unit change in the NINO3.4 index, but can also be applied to historical NINO3.4 values to estimate the change in cases due to El Niño events. Here, we focused on the four strongest El Niño events of 1982–83, 1997–98, 2015–16, and 2023–24. We defined counterfactual ENSO wherein El Niño and La Niña events did not occur by setting the corresponding positive and negative NINO3.4 values to zero. We then used the Delta method[31,76] to calculate the case change rate [$\Delta \text{case}_{i(t+L)}$] in the occurrence year ($L = 0$), following year ($L = 1$), and the second year ($L = 2$) due to the El Niño event via temperature and precipitation:

$$\Delta \text{case}_{i(t+L)} = \exp \left( (\theta_L^{temp} \text{deNINO3.4}_t^0 * \tau_i^{temp} + \theta_L^{precip} \text{deNINO3.4}_t^0 * \tau_i^{precip}) \right.$$
$$\left. - (\theta_L^{temp} \text{deNINO3.4}_t * \tau_i^{temp} + \theta_L^{precip} \text{deNINO3.4}_t * \tau_i^{precip}) \right) - 1 \quad (5)$$

where $deNINO3.4_t$ represents the observed NINO3.4 value in the year of the El Niño event ($t$), and $\text{deNINO3.4}_t^0$ represents the counterfactual ENSO (zero) in that year. We added this case change rate to the observed cases in those three lagged years ($L = 0, 1, 2$) for each country, obtaining the change from counterfactual (without the event) to observed (with the event) cases. The cumulative effects of each El

Niño event were calculated as the sum of changes in three lagged years after the event. Global case changes due to the event were then calculated as the sum of the changes in dengue cases in 57 countries. Note that our estimate of the global change in cases attributable to the 2023–24 El Niño was based solely on the observed cases in 2024, the year of occurrence, and that we would expect this estimate to increase when dengue case data for 2025 and 2026 become available.

## CMIP model selection

We used multiple ensemble members of CMIP6 climate models that are available under four emission scenarios (SSP1-2.6, SSP2-4.5, SSP3-7.0, and SSP5-8.5) to predict changes in dengue cases due to ENSO evolution under future warming. We tested 88 multi-model ensemble members for SSP1-2.6, 150 for SSP2-4.5, 134 for SSP3-7.0, and 94 for SSP5-8.5. We measured ENSO amplitude as the standard deviation of the NINO3.4 time series[45]. We also assessed the frequency of ENSO events but found a high uncertainty with great variations across multi-model ensemble members and a large bias between simulated and observed frequencies of ENSO events during 1980–2019. Thus, we chose the relatively robust ENSO amplitude to predict dengue cases.

To ensure that our projections are feasible, we defined "skilful" ensemble members as the absolute value of the bias between simulated and observed ENSO amplitude during 1980–2019 is less than 50% of the observed ENSO amplitude[77]. Based on those selected skilful ensemble members, we calculated the population-weighted monthly temperature and daily precipitation to assess the teleconnections for each country during 1940–2019 and 2020–2099. The change rates in ENSO amplitude and teleconnection strength (the absolute value of teleconnection) were obtained to project changes in dengue cases in the next section.

## Projecting future effects of ENSO

To estimate the impact of projected future changes in ENSO on dengue cases, we compared scenarios where ENSO is projected to change with a counterfactual where it remains the same as over the observed period (1980–2019). We defined the observed ENSO and teleconnections as the "counterfactual future", and the "future" ENSO and teleconnections were calculated by adding the change rate between the historical and future simulations to the observed values (Eqs. 6–7). This method ensures that the "counterfactual future" retains the same amplitude and teleconnections as the observed ENSO from the historical period. As such, the change in cases between the "counterfactual future" and "future" represents the dengue cases associated with projected changes in ENSO evolution under different warming scenarios in the 21st century[31,78].

$$deNINO3.4_F^e = deNINO3.4_{CF}^e * (1 + \%\text{change in ENSO amplitude}) \quad (6)$$

$$\tau_{i,F}^c = \tau_{i,CF}^c * (1 + \%\text{change in teleconnection strength}_i^c) \quad (7)$$

We then estimated the change rate in dengue cases due to future ENSO changes in the scenarios of interest (SSP1-2.6, SSP2-4.5, SSP3-7.0, SSP5-8.5) as

$$\Delta case_{i(t+L)}^e = \exp\left(\sum_{L=0}^{j}\left(\left(\theta_L^{temp} deNINO3.4_F^e * \tau_{i,F}^{temp} + \theta_L^{precip} deNINO3.4_F^e * \tau_{i,F}^{precip}\right) - \left(\theta_L^{temp} deNINO3.4_{CF}^e * \tau_{i,CF}^{temp} + \theta_L^{precip} deNINO3.4_{CF}^e * \tau_{i,CF}^{precip}\right)\right)\right) - 1 \quad (8)$$

where $deNINO3.4_F^e$ and $deNINO3.4_{CF}^e$ represent the future and counterfactual future ENSO time series, respectively. $\tau_{i,F}^c$ and $\tau_{i,CF}^c$ represent the future and counterfactual future teleconnections for each country $i$, respectively. $c$ denotes the local climatic variables. We projected the country-level change rate in dengue cases for multi-

model ensemble members, using 1000 bootstrap estimates of our empirical model. The average change rate in global dengue cases due to changes in ENSO amplitude and teleconnections between future and counterfactual future was calculated as the mean of country-level change rates. We also estimated the average change rate in global dengue cases due to changes in El Niño and La Niña events between future and counterfactual future, based on the average detrended NINO3.4 value during the past El Niño and La Niña events separately, where $e$ denotes the ENSO events.

## Uncertainty in the nonlinear association between future warming and cases

Given that the optimal temperature for dengue transmission is estimated to be around 29 °C[48], we expected that our predictions might be less powerful for the countries with temperatures exceeding 29 °C during 2020–2099. Here, we used the monthly average temperature for each country during 2020–2099 to assess the power of our predictions of dengue risk based on our linear regression model.

We first corrected the CMIP-predicted monthly average temperatures from 2020 to 2099, based on the bias between the observed temperatures and CMIP-predicted temperature for each country and month over 1980–2019:

$$T_{imt} = T_{imt}^{CMIP} + (\widetilde{T}_{im}^{observed} - \widetilde{T}_{im}^{CMIP}) \quad (9)$$

where $T_{imt}$ represents the corrected monthly average temperature, $T_{imt}^{CMIP}$ represents the monthly average temperature predicted from each CMIP6 simulation in country $i$ in month $m$ and year $t$, $\widetilde{T}_{im}^{observed}$ denotes the observed temperature in country $i$ in month $m$ over 1980–2019, and $\widetilde{T}_{im}^{CMIP}$ denotes the temperature predicted from this CMIP model over the same period. This correction largely reduced the Root Mean Squared Error (RMSE) from ~4.5 to ~3.5 between observed and CMIP-predicted temperatures over 1980–2019. We then calculated the proportion of countries with average summertime temperatures and annual maximum temperatures during 2020–2099 exceeding 29 °C to represent the uncertainty in our projections of rising dengue risk.

## Reporting summary

Further information on research design is available in the Nature Portfolio Reporting Summary linked to this article.

# Data availability

The dengue case data are deposited in the OpenDengue database [https://opendengue.org/]. The climate index data are available from Climate index dataset of NCC and El Niño Index Dashboard of NOAA. Local temperature and precipitation data are available from ERA5 [https://www.ecmwf.int/en/forecasts/datasets/reanalysis-datasets/era5]. The population data are available from GPW version 4 and World Population Prospects 2024 [https://population.un.org/wpp/Download/Standard/Population/]. The CMIP6 ensembles are openly available via PCMDI/LLNL (California) [https://pcmdi.llnl.gov/CMIP6/]. Reported dengue cases for 2023 and 2024 in sensitivity analysis are available upon request to the corresponding author and with permission from the data provider (Huaiyu Tian and Oliver J. Brady). The request will be responded within 2 weeks. Source data are provided with this paper on the following GitHub repository: https://github.com/huaiyutian/Dengue_ENSO. Source data are provided with this paper.

# Code availability

Code files to generate all the figures in the article are available on the following GitHub repository: https://github.com/huaiyutian/Dengue_ENSO.

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

## Acknowledgements

We are deeply grateful to Dr. C. Jessica E. Metcalf for her valuable comments on this manuscript. This study was supported by the Fundamental Research Funds for the Central Universities (2233300001); Beijing Research Center for Respiratory Infectious Diseases Project (BJRID2025-001); National Key Research and Development Program of China (2022YFC2303803); Scientific and technological innovation 2030—major project of new generation artificial intelligence (2021ZD0111201); Science and Technology Projects of Xizang Autonomous Region, China (XZ202501JD0012); Key research projects of Beijing Natural Science Foundation-Haidian District Joint Fund (L232014). O.J.B. acknowledges support from the UK Medical Research Council grants MR/V031112/1 (which also supports A.L.) and MR/Y004663/1 (which also supports A.M.C.). O.J.B., K.J., and K.M.S. also acknowledge support from the AXA Research Foundation (grant: "Pioneering a global observatory for dengue outbreak early warning"). The funders had no role in study design, data collection and analysis, the decision to publish, or in preparation of the manuscript.

## Author contributions

H.T. conceived the study. H.T. and O.J.B. jointly supervised this work. Y.X., Y.L., Z.Z., K.M.S., Y.C., K.J., A.C., A.L., and Y.W. collected the statistical data. Y.T., Y.X., and Y.L. conducted the analyses. Y.T., O.J.B., and H.T. edited the manuscript. Y.T., Y.X., Y.L., and H.T. wrote the manuscript. Y.C., Q.L., Z.M., C.S., J.F., Q.H., Z.W., Y.Y., Y.G., K.L., and B.C. provided important comments on the draft manuscript and edited the manuscript. All authors read and approved the manuscript.

## Competing interests

The authors declare no competing interests.

## Additional information

¹National Key Laboratory of Intelligent Tracking and Forecasting for Infectious Diseases, Joint International Research Laboratory of Catastrophe Simulation and Systemic Risk Governance, School of National Safety and Emergency Management, Beijing Normal University, Beijing, China. ²State Key Laboratory of Remote Sensing and Digital Earth, Beijing Key Laboratory of Surveillance, Early Warning and Pathogen Research on Emerging Infectious Diseases, Beijing Research Center for Respiratory Infectious Diseases, Center for Global Change and Public Health, Beijing Normal University, Beijing, China. ³Centre for the Mathematical Modelling of Infectious Diseases, London School of Hygiene & Tropical Medicine, London, UK. ⁴Department of Infectious Disease Epidemiology and Dynamics, Faculty of Epidemiology and Population Health, London School of Hygiene & Tropical Medicine, London, UK. ⁵National Engineering Research Center of Eco-Environment in the Yangtze River Economic Belt, China Three Gorges Corporation, Wuhan, China. ⁶Department of Epidemiology, Johns Hopkins Bloomberg School of Public Health, Baltimore, MD, USA. ⁷College of Letters & Science, University of Wisconsin-Madison, Madison, WI, USA. ⁸Department of Biological Sciences, Purdue University, West Lafayette, IN, USA. ⁹Institut de Biologie de l'École Normale Supérieure UMR 8197, Eco-Evolutionary Mathematics, École Normale Supérieure, Paris, France. ¹⁰Unité Mixte Internationnale 209, Mathematical and Computational Modeling of Complex Systems, Sorbonne Université, Paris, France. ¹¹School of Computer Science and Engineering, Beihang University, Beijing, China. ¹²These authors contributed equally: Yunyu Tian, Yiting Xu, Yilin Liang. ✉e-mail: Oliver.Brady@lshtm.ac.uk; tianhuaiyu@gmail.com

