## [Peer Review file · Nature Communications]

Rising dengue risk with increasing El Niño–Southern Oscillation amplitude and teleconnections

Corresponding Author: Professor Huaiyu Tian

Version 0:

Reviewer comments:

Reviewer #1

(Remarks to the Author)

The manuscript is well-structured, and the methods are consistent with the objectives and motivations outlined in the introduction. The results present a global analysis of the impacts of ENSO on dengue cases and outbreaks across 57 countries. While the association between ENSO and dengue has been explored in the literature, some aspects of this work represent a valuable and original contribution, particularly the modeling of dengue cases using ENSO indices at a global scale—whereas most previous studies tend to focus on individual countries.

Although the global perspective is one of the paper's key strengths, it also presents certain limitations. This broad approach does not account for regional differences in ENSO impacts within large countries, such as Brazil, where different regions experience opposite responses to El Niño and La Niña events. By aggregating data at the national level, some of this spatial heterogeneity may be lost. Nevertheless, this does not detract from the authors' ability to meet their objectives; the study successfully identifies general patterns of ENSO influence on dengue burden across countries and provides quantifiable results that are meaningful for the scientific and public health community. The discussion section addresses most of the limitations and potential sources of uncertainty, making the work transparent and demonstrating a strong awareness of the complexity of dengue transmission.

A few minor revisions are needed to improve clarity in the presentation of results and methods, and to correct some textual inconsistencies. For example, line 75 refers to "other similar indices such as Indian Ocean basin-wide," yet the manuscript does not return to or incorporate any such indices. It would be helpful to clarify that other modes of climate variability — such as the Indian Ocean Dipole (IOD) and the Madden–Julian Oscillation (MJO) — exist, and to explain the rationale for focusing exclusively on ENSO. Without such clarification, a non-expert reader may incorrectly assume that ENSO is the only relevant climate mode.

Summarized, the main issues are:

-Regional ENSO effects

The paper currently treats ENSO impacts at a national scale, but in large countries like Brazil, ENSO has opposing effects across regions. For example, El Niño tends to suppress rainfall in the northeast but enhances it in the southeast. This spatial heterogeneity should be acknowledged to avoid overgeneralization (see Methods section, e.g., line 521).

-Clarification on climate modes

Line 75 references "other similar indices such as Indian Ocean basin-wide," but no further mention or analysis is provided. To avoid confusion, either remove this mention or explain why only ENSO was considered. A brief acknowledgment of other relevant climate modes (e.g., Indian Ocean Dipole, Madden–Julian Oscillation) would improve the conceptual framing.

-Link between teleconnections and climate change

Clarify that teleconnections, such as those associated with ENSO, occur independently of climate change. It would be helpful to reframe the discussion to emphasize that understanding teleconnections helps anticipate dengue outbreaks, regardless of long-term climate trends.

-Improved data visualization

Instead of only using tables to present data availability by country, consider adding a world map highlighting countries with full vs. partial data. This would make spatial patterns more intuitive and enhance reader comprehension.

-Niño indices Visualization

In the figures, the Niño indices use similar blue color shades, making it difficult to distinguish them (e.g., Niño 3, 3.4, MEI). Use more contrasting colors to improve figure readability.

In addition to the general comments above, I have provided detailed, line-by-line suggestions in the attached PDF. These include minor textual corrections, clarifications on climate variability concepts, and specific suggestions to improve data presentation and readability.

Overall, this is a strong and well-executed study. Despite a few minor issues, all parts of the manuscript are well-integrated. The methodology is particularly solid and appropriate for the study's goals, which are convincingly achieved.

(Remarks on code availability)

The code is fully open source and written in Python, using well-known libraries such as pandas, xarray, and numpy. It is easily reproducible, as most of the data used are open access and included or referenced in the code. The repository also includes a README file with sufficient instructions for use and adaptation in future studies.

Reviewer #2

(Remarks to the Author)

The authors present an analysis of past and future impacts of El Niño on regional climates and hence dengue cases on a (nearly) global scale. While the effect of El Niño on vector borne diseases has been investigated for several regions or countries individually, this study bundles these findings together and explains them with the magnitude of teleconnections. The methodology seems robust, very comprehensive, and reproducible (with only a few clarifications needed).

Main comments

The authors need to define what they exactly mean by "dengue risk". Does it simply mean dengue case numbers? An increase of an undefined risk by 48% (1.181) would not make sense.

El Niño impacts the local climate/weather on the African continent which has significant dengue transmission too. But this is not included in this study at all and the reasons for this/implications need at least discussion.

Minor comments

Title

The authors focus solely on El Niño, maybe this should be reflected in the title?

Intro

I do not feel the term "teleconnections" is commonly used - it might be best to give a brief description at the beginning. l.75 missing a word?

Methods

eq 1: typo,  =

CMIP model selection - how do you define skillful?

Results

Fig 1b is not easy to interpret - how do you explain this linear trend?

l.204 - why "mainly"? trade-off and synergy covers 100% of the area. It might be best to describe this figure in more detail.

(Remarks on code availability)

I briefly checked the Python code which seems to be doing what it should do. The full required data needs to be requested/downloaded from other sources but that is understandable.

There is a README with enough instructions.

Reviewer #3

(Remarks to the Author)

Dear Editors and Authors, thank you for the opportunity to review this study that explored the impact of ENSO on dengue fever. The manuscript is well drafted. Some comments for the authors to consider as attached. Hopefully, it can be helpful for further improvement.

Introduction:

Line 81: It would be beneficial to further elaborate on the concept of teleconnection: What does it entail, and how does it contribute to the association between ENSO and dengue epidemics/incidence?

Methods:

Line 459: This line states that data for the year 2024 were unavailable, and line 461 notes that data from 2020 to 2024 were not included. Does this mean that only dengue data from 1980 to 2019 were used in the analysis?

Line 729: This line indicates that dengue data for 2023 and 2024 are available upon reasonable request. If these data were excluded from the analysis, what is the rationale for mentioning their availability? Please clarify.

Line 257 (Figure 3): Regarding the legend in Figure 3a: How can the reader identify areas marked as "not significant"? Are these limited to the small area along the China–India border? Please clarify.

Results:

Line 248: The manuscript states that the greater effects of the 2023–2024 events were likely due to a strong El Niño. However, since the 2023–2024 dengue data were excluded from the modeling, does this statement refer to a simulated effect or is it based on other sources of evidence? Please clarify.

(Remarks on code availability)

Version 1:

Reviewer comments:

Reviewer #1

(Remarks to the Author)

I find that the authors have addressed my previous comments satisfactorily, and the manuscript is now considerably clearer and more coherent, with many of my concerns appropriately addressed.

However, I would like to point out a conceptual inaccuracy in the definition of teleconnections in Response 1.6.5, where it is stated that "... the former process has been thoroughly investigated in climatology as teleconnections which is defined as the correlation between ENSO and local climate," which is not entirely correct. In climatology, teleconnections are not merely statistical correlations, but rather physical atmospheric mechanisms that allow climatic anomalies in one region to influence distant areas. A well-documented example is the impact of ENSO on the Brazilian climate through shifts in the Walker circulation and the propagation of Rossby waves. I recommend rephrasing this section to better reflect the physical and dynamical basis of teleconnections.

Aside from this point, all my other concerns have been properly addressed, including the issues related to the figures, which are now more intuitive and clearer than before.

(Remarks on code availability)

Reviewer #2

(Remarks to the Author)

The authors have addressed all the reviewer's comments satisfyingly. I have got nothing further to add.

(Remarks on code availability)

Reviewer #3

(Remarks to the Author)

Dear Editors and Authors, thank you for the opportunity to review the revised manuscript. My concerns have been well addressed. Congratulations to the authors!

(Remarks on code availability)

Reviewer 1:

Please note: The reviewer's comments are presented in regular text. The responses are in blue. Page and line numbers refer to those in the cleaned manuscript. Revised sentences in the updated manuscript are in *italics* and new sentences are underlined, presented at the end of each corresponding response.

General Comments:

The manuscript is well-structured, and the methods are consistent with the objectives and motivations outlined in the introduction. The results present a global analysis of the impacts of ENSO on dengue cases and outbreaks across 57 countries. While the association between ENSO and dengue has been explored in the literature, some aspects of this work represent a valuable and original contribution, particularly the modeling of dengue cases using ENSO indices at a global scale—whereas most previous studies tend to focus on individual countries.

Although the global perspective is one of the paper's key strengths, it also presents certain limitations. This broad approach does not account for regional differences in ENSO impacts within large countries, such as Brazil, where different regions experience opposite responses to El Niño and La Niña events. By aggregating data at the national level, some of this spatial heterogeneity may be lost. Nevertheless, this does not detract from the authors' ability to meet their objectives; the study successfully identifies general patterns of ENSO influence on dengue burden across countries and provides quantifiable results that are meaningful for the scientific and public health community. The discussion section addresses most of the limitations and potential sources of uncertainty, making the work transparent and demonstrating a strong awareness of the complexity of dengue transmission.

A few minor revisions are needed to improve clarity in the presentation of results and methods, and to correct some textual inconsistencies. For example, line 75 refers to "other similar indices such as Indian Ocean basin-wide," yet the manuscript does not return to or incorporate any such indices. It would be helpful to clarify that other modes of climate variability – such as the Indian Ocean Dipole (IOD) and the Madden Julian Oscillation (MJO) – exist, and to explain the rationale for focusing exclusively on ENSO. Without such clarification, a non-expert reader may incorrectly assume that ENSO is the only relevant climate mode.

Response to General Comments: We express our gratitude to the reviewer for providing positive and constructive comments on our manuscript. We have addressed the minor revisions point by point and clarified other climate modes relevant to global climate variability. Please see our specific responses below.

Comment 1.1:

-Regional ENSO effects

The paper currently treats ENSO impacts at a national scale, but in large countries like Brazil, ENSO has opposing effects across regions. For example, El Niño tends to suppress rainfall in the northeast but enhances it in the southeast. This spatial

heterogeneity should be acknowledged to avoid overgeneralization (see Methods section, e.g., line 521).

Response 1.1: We sincerely appreciate this insightful comment. We agree that the teleconnection between ENSO and local climate may vary within a country. In this study, we calculated country-level teleconnection to match the country-level disease data and evaluate the general effects of ENSO on country-level cases. We have now clarified the potential spatial heterogeneity in the Methods section and added the recommendation for city-level research in the Discussion section.

Lines 542–551 in Methods (Revised): *“To match the country-level disease data resolution, we measured ENSO teleconnections as the extent to which each country’s climate is influenced by ENSO, accounting for the effects of temperature and precipitation, and different time scales at which teleconnections may manifest. Unlike previous approaches that examine the impacts of global climate and local climate on dengue epidemics separately, this method allows for a comprehensive study of the impact of ENSO-driven local climate on dengue epidemics, providing a more nuanced understanding of their relationships. Although ENSO may have opposing effects within large countries like Brazil⁶⁹, this study focuses on assessing the long-term effects of ENSO on country-level dengue cases due to data availability.”*

Lines 440–444 in Discussion (New): *“Third, our country-level teleconnections may overlook subnational heterogeneity in ENSO-dengue relationships particularly for large countries. Future studies with access to higher-resolution case data could build upon our findings by examining these subnational patterns such as city-level associations.”*

Comment 1.2:

Clarification on climate modes

Line 75 references “other similar indices such as Indian Ocean basin-wide,” but no further mention or analysis is provided. To avoid confusion, either remove this mention or explain why only ENSO was considered. A brief acknowledgment of other relevant climate modes (e.g., Indian Ocean Dipole, Madden-Julian Oscillation) would improve the conceptual framing.

Response 1.2: Thanks for raising this important point. We acknowledge that various climate modes are interacted with each other and similarly reflect global climate variability. We have now extended other climate modes and clarified our focus on ENSO which is the most prominent and well-known indicator of global climate variability based on previous studies.

Lines 77–80 in Introduction (Revised): *“Given the extensive body of research, we hereafter use El Niño – Southern Oscillation (ENSO) as the well-known indicator of global climate variability, among other similar indices such as the Indian Ocean Dipole (IOD) and the Madden Julian Oscillation (MJO), to investigate this question for simplification.”*

Comment 1.3:

-Link between teleconnections and climate change

Clarify that teleconnections, such as those associated with ENSO, occur independently of climate change. It would be helpful to reframe the discussion to emphasize that understanding teleconnections helps anticipate dengue outbreaks, regardless of long-term climate trends.

Response 1.3: Thanks for your great suggestion. In the calculation of teleconnections, we conducted a linear detrend algorithm and z-score standardisation to remove the effects of long-term warming trends. We have now emphasized the importance of ENSO teleconnections on anticipating dengue outbreaks in the Discussion section.

Lines 364–368 in Discussion (Revised): *“As ENSO teleconnections we assessed are independent of long-term climate change, our new understanding based on teleconnections can be used to develop medium- and long-range outbreak forecasting systems to prevent and mitigate dengue outbreaks, and to empower studies that attribute the effects of climate change on current epidemics.”*

Lines 419–421 in Discussion (Revised): *“Future prospects highlight the unrecognized impact of global climate variability on the magnitude of dengue epidemics, which is independent of long-term warming trends.”*

Comment 1.4:

-Improved data visualization

Instead of only using tables to present data availability by country, consider adding a world map highlighting countries with full vs. partial data. This would make spatial patterns more intuitive and enhance reader comprehension.

Response 1.4: Thanks for your great suggestion. We have added a world map with color coding to distinguish countries with and without data availability in the Supplementary Information.

Lines 467–468 in Methods (Revised): *“Annual reported dengue cases used in this study were compiled from 57 countries covering the period from 1980 to 2024 (Supplementary Fig. 13 and Table 5).”*

Line 166 in Supplementary Information (New):

Study Area: 57 Countries with Reported Dengue Cases (1980–2024)

Supplementary Fig. 13 Map of dengue case data availability from 1980 to 2024.

Comment 1.5:

-Niño indices visualization

In the figures, the Niño indices use similar blue color shades, making it difficult to distinguish them (e.g., Niño 3, 3.4, MEI). Use more contrasting colors to improve figure readability.

Response 1.5: Thanks for your great suggestion on figures. We have now adjusted the color schemes and distinguished NINO3, NINO3.4, and MEI for Supplementary Fig. 1a.

Line 12 in Supplementary Information (Revised):

Comment 1.6:

In addition to the general comments above, I have provided detailed, line-by-line suggestions in the attached PDF. These include minor textual corrections, clarifications on climate variability concepts, and specific suggestions to improve data presentation and readability.

Response 1.6: Thanks for your detailed comments on the manuscript. We have now addressed the comments point by point.

Comment 1.6.1:

Line 59: There are, especially vaccines, but since dengue is considered a neglected tropical disease, it continues to be a frequent problem, mostly due to vector transmission and the lack of effective mitigation policies.

Response 1.6.1: Thanks for providing this important insight on the current dengue status. We agree that while dengue vaccines represent important progress, their limited global implementation and variable efficacy profiles, combined with persistent challenges in vector control and insufficient public health policies, continue to sustain dengue's status as a neglected tropical disease and growing public health threat. We have now clarified this in the Introduction section.

Lines 62–64 in Introduction (Revised): *“While dengue vaccines have recently made important progress^{8,9}, dengue continues to pose an escalating threat to public health due to persistent vector transmission and the lack of vaccine availability and effective mitigation policies.”*

Comment 1.6.2, 1.6.3, and 1.6.4:

Line 66: It is true that El Niño affects all of these regions, but the manner and intensity of its impact vary significantly. El Niño is much more strongly correlated with countries like Peru and Ecuador, for example, as it occurs near or very close to their coastlines, resulting in a direct ocean-atmosphere interaction. In contrast, for regions like Sri Lanka – mentioned in the text – the effects are much more indirect and rely on teleconnection mechanisms to propagate across the globe and exert influence over such long distances. The Indian Ocean Dipole (IOD) has a more direct correlation with Sri Lanka and Asia than ENSO, although interactions between ENSO and the IOD can occur. ENSO is not the only, nor necessarily the main, influencing factor.

Line 71: As mentioned in the comment above.

Line 75: If you are considering this, it should be spoken about IOD and even Madden-Julian Oscillation (MJO) as the same level of importance as ENSO.

Response 1.6.2, 1.6.3, and 1.6.4: Thanks for your comment on the spatial-varying effects of ENSO and pantropical climate interactions. We acknowledge that ENSO influences country-level climate in different magnitudes according to the distance. Thus, we used teleconnections to measure the magnitude of the association between ENSO and country-level climate. As we are aware of the interactions between climate modes such as ENSO, IOD, and MJO, we selected ENSO as the most prominent and well-known indicator of global climate variability to explore the mechanism of the impact of global climate variability on global dengue epidemics. We have now elaborated this in the Introduction section.

Lines 77–80 in Introduction (Revised): *“Given the extensive body of research, we hereafter use El Niño – Southern Oscillation (ENSO) as the well-known indicator of global climate variability, among other similar indices such as the Indian Ocean Dipole*

(IOD) and the Madden Julian Oscillation (MJO), to investigate this question for simplification.”

Comment 1.6.5:

Line 86: I do not see a direct link between teleconnections and climate change impacts. Teleconnections occur independently of climate change—they have always existed and will continue to exist, as the atmosphere is a fluid system that constantly redistributes energy across the globe. Perhaps you could focus instead on the idea that a better understanding of the impacts of teleconnections on dengue outbreaks could help to anticipate future outbreaks and explain past ones, as you begin to discuss in the following paragraph.

Response 1.6.5: Thanks for your great suggestion. We elaborated the ENSO-climate-dengue link in this paragraph and focused on the impacts of teleconnections on dengue epidemics in the following paragraph. We think that clarifying the ENSO-climate-dengue link in terms of two processes is necessary before emphasizing the role of teleconnections on dengue epidemics. We have now adjusted this paragraph to clarify the link from ENSO to dengue epidemics through teleconnections.

Lines 82–88 in Introduction (Revised): *“Manifested as sea surface temperature anomalies in the tropical Pacific Ocean, ENSO influences dengue risk through “teleconnections” – ENSO-induced atmospheric circulation alters local climates that affect mosquito ecology^{25,26}. Although the former process has been thoroughly investigated in climatology as teleconnections which is defined as the correlation between ENSO and local climate^{27,28}, an open question remains regarding the latter process; namely, how do teleconnections contribute to the different associations between ENSO and dengue epidemics²⁹?”*

Comment 1.6.6:

Line 96: Exactly, that is the right path.

Response 1.6.6: Thanks for your positive comment.

Comment 1.6.7:

Line 103: Climate change becomes relevant to this topic primarily by creating more suitable environments for disease transmission, mainly due to the rise in global average temperatures, rather than through any direct interaction with ENSO or other modes of climate variability.

Response 1.6.7: Thanks for your concern. We agree that climate change primarily exacerbates dengue risk by expanding thermally suitable environments for disease transmission. Here we intended to state the increasing ENSO variability under future emission scenarios based on the previous research (Cai et al., 2022) to emphasize the need to investigate the future effects of ENSO on dengue risk. We have now replaced “climate change” by “future emission scenarios” for clarification.

Lines 106–109 in Introduction (Revised): “With increasing ENSO variability in recent years and growing evidence that variability will further increase under future emission scenarios^{31,39}, there is an urgent need to investigate how ENSO shapes future risk of dengue^{40,41}.”

Reference:

Cai, W. et al. Increased ENSO sea surface temperature variability under four IPCC emission scenarios. *Nat Clim Chang* 12, 228–231 (2022).

Comment 1.6.8:

Line 107: If no data related to the Indian Ocean Dipole (IOD) were used in your analysis, I would suggest removing the reference to it in line 75, where you mention 'other similar indices such as the Indian Ocean basin-wide.' This reference may give the impression that IOD was considered, while the text suggests the focus is solely on ENSO indices.

Response 1.6.8: Thanks for your suggestion. We did not use IOD in our analysis. We have now removed the reference in line 75.

Comment 1.6.9:

Line 121: The Niño indices use very similar color schemes, which may make it difficult for readers to distinguish between them. I would suggest using more distinct colors, particularly for indices like Niño 3, 3.4, and MEI, to improve clarity.

Response 1.6.9: Thanks for your suggestion on visualization. We have now adjusted the color schemes for Supplementary Fig. 1a.

Line 12 in Supplementary Information (Revised):

Comment 1.6.10:

Line 124: What countries exactly? are they more in America or Asia? Please specify.

Response 1.6.10: Thanks for your question. The countries with significant correlations and the greatest total number of cases between 1980 and 2019 – Brazil, Vietnam, and

Indonesia – are distributed in South America and Asia. We have now labeled these countries in Fig. 2b and described them in the Results section.

Lines 126–133 in Results (Revised): *“Although long-term correlations between the time series of ENSO indices and annual dengue cases are insignificant in most countries, there exists a great magnitude of dengue epidemics in endemic countries such as Brazil, Vietnam, and Indonesia strongly affected by El Niño, i.e., more cases during El Niño years but fewer cases during La Niña years, as evidenced by both the linear trend and the clustering of endemic countries above the one-to-one line in Fig. 1b.”*

Comment 1.6.11:

Line 143: Within these countries, the associations become even more complex. For example, in Brazil, different regions exhibit distinct patterns of precipitation and temperature anomalies in response to El Niño events, as mentioned in the methods section (line 521).

Response 1.6.11: Thanks for your question. We agree that the associations between ENSO and local climate can be more complex within a country. In this study, we calculated country-level teleconnection to match the country-level disease data and evaluate the general effects of ENSO on country-level cases. We have now clarified the potential spatial heterogeneity in the Methods section.

Lines 542–551 in Methods (Revised): *“To match the country-level disease data resolution, we measured ENSO teleconnections as the extent to which each country’s climate is influenced by ENSO, accounting for the effects of temperature and precipitation, and different time scales at which teleconnections may manifest. Unlike previous approaches that examine the impacts of global climate and local climate on dengue epidemics separately, this method allows for a comprehensive study of the impact of ENSO-driven local climate on dengue epidemics, providing a more nuanced understanding of their relationships. Although ENSO may have opposing effects within*

large countries like Brazil⁶⁹, this study focuses on assessing the long-term effects of ENSO on country-level dengue cases due to data availability.

Comment 1.6.12:

Line 229: Excellent, but I found this type of description missing from the earlier results. It is important to show how these countries are spatially distributed – simply stating the number (e.g., in lines 137 and 142) does not allow the reader to identify any spatial patterns.

Response 1.6.12: Thanks for your comment. We summarized the number and listed some endemic countries of trade-off and synergy types in the earlier results (e.g., in lines 143 and 148 in the revised manuscript), and then we analyzed the spatial pattern of the two types in Figure 3. We believe that Figure 3 shows readers a complete spatial pattern of ENSO effects and clear regional comparison across continents and climate divisions.

Comment 1.6.13:

Line 433: Yes, very good.

Response 1.6.13: Thanks for your positive comment.

Comment 1.6.14:

Line 444: The methods are clearly explained and align well with the stated objectives. However, there is no mention of the Indian Ocean basin in the methodology or analysis. I would suggest removing the reference to it in the introduction, as the study appears to focus primarily on the influence of ENSO – which is entirely appropriate for the scope of this work.

Response 1.6.14: Thanks again for your positive comment. We did not use IOD in our analysis. We have now removed the reference in line 75.

Comment 1.6.15:

Line 447: Instead of a table, I suggest make a world map with the countries, with color coding to distinguish between those with full data availability and those without. This approach may offer a clearer and more practical visualization for readers.

Response 1.6.15: Thanks for your practical suggestion on visualization. We have added a world map with color coding to distinguish countries with and without data availability in the Supplementary Information.

Lines 467–468 in Methods (Revised): *“Annual reported dengue cases used in this study were compiled from 57 countries covering the period from 1980 to 2024 (Supplementary Fig. 13 and Table 5).”*

Line 166 in Supplementary Information (New):

Study Area: 57 Countries with Reported Dengue Cases (1980–2024)

Supplementary Fig. 13 Map of dengue case data availability from 1980 to 2024.

Comment 1.6.16:

Line 521: I'm not entirely convinced by this point. For instance, Brazil has several regions that respond differently to ENSO events: in the Northeast, La Niña tends to increase precipitation while El Niño decreases it, whereas the opposite pattern is generally observed in the Brazilian's Southeast region (see <https://doi.org/10.1111/nyas.14592>). It is not clear how this type of regional heterogeneity is accounted for in the proposed method.

Response 1.6.16: Thanks for your concern and providing this reference. We agree that the associations between ENSO and local climate can be more complex within a country. In this study, we calculated country-level teleconnection to match the country-level disease data regardless of regional heterogeneity within countries. We have now clarified the potential spatial heterogeneity in the Methods section.

Lines 542–551 in Methods (Revised): *“To match the country-level disease data resolution, we measured ENSO teleconnections as the extent to which each country's climate is influenced by ENSO, accounting for the effects of temperature and precipitation, and different time scales at which teleconnections may manifest. Unlike previous approaches that examine the impacts of global climate and local climate on dengue epidemics separately, this method allows for a comprehensive study of the impact of ENSO-driven local climate on dengue epidemics, providing a more nuanced understanding of their relationships. Although ENSO may have opposing effects within large countries like Brazil⁶⁹, this study focuses on assessing the long-term effects of ENSO on country-level dengue cases due to data availability.”*

Comment 1.6.17:

Line 527: correct

Response 1.6.17: Thanks for your encouragement.

Comment 1.6.18:

Line 572: that is a valid assumption: dengue cases refer to individuals who experience symptoms and seek treatment at a healthcare facility. However, prior to the onset of symptoms, there is often a significant lag, as numerous events involving the vector occur, potentially leading to a delay between the climate event that triggered vector breeding and the manifestation of symptoms. One detail that may have been overlooked is the temporal frequency, which is yearly rather than monthly. Nevertheless, I believe that, for the purposes of your objectives, this approach is appropriate.

Response 1.6.18: We appreciate this thoughtful discussion. We fully agree that monthly-scale analyses would provide more precise insights into the lagged effects between climatic drivers and disease manifestation, given the complex cascade of vector-related events preceding symptom onset. While we did collect monthly dengue case reports, the limited temporal coverage of this global dataset (2014–2023) constrained its utility for our objectives. We share your hope that future efforts will generate longer-term, high-resolution datasets to enable robust time-lag analyses. Your perspective has enriched our understanding of the lagged effects, and we sincerely appreciate your engagement.

General Comments:

Overall, this is a strong and well-executed study. Despite a few minor issues, all parts of the manuscript are well-integrated. The methodology is particularly solid and appropriate for the study's goals, which are convincingly achieved.

The code is fully open source and written in Python, using well-known libraries such as pandas, xarray, and numpy. It is easily reproducible, as most of the data used are open access and included or referenced in the code. The repository also includes a README file with sufficient instructions for use and adaptation in future studies.

Response to General Comments: We sincerely appreciate the reviewer's positive comments and constructive suggestions on our initial submission and the confirmation of the code files. By adding clarification and adjusting visualization, we believe that the manuscript has been largely improved.

Reviewer 2:

Please note: The reviewer's comments are presented in regular text. The responses are in blue. Page and line numbers refer to those in the cleaned manuscript. Revised sentences in the updated manuscript are in *italics* and new sentences are underlined, presented at the end of each corresponding response.

General comments:

The authors present an analysis of past and future impacts of El Niño on regional climates and hence dengue cases on a (nearly) global scale. While the effect of El Niño on vector borne diseases has been investigated for several regions or countries individually, this study bundles these findings together and explains them with the magnitude of teleconnections.

The methodology seems robust, very comprehensive, and reproducible (with only a few clarifications needed).

Response to General Comments: We express our gratitude to the reviewer for providing positive and insightful comments on our manuscript.

Comment 2.1:

The authors need to define what they exactly mean by "dengue risk". Does it simply mean dengue case numbers? An increase of an undefined risk by 48% (1.181) would not make sense.

Response 2.1: Thanks for your important concern about the "dengue risk". We refer "dengue risk" to annual dengue cases estimated by our models. We have now clarified the definition of dengue risk in this study and replaced "dengue risk" by "annual dengue cases" in the Results section.

Lines 184–193 in Results (Revised): *"ENSO primarily increased dengue risk (represented by annual dengue cases) via rising local temperature. Conversely, its impact via local precipitation was smaller, and it varied in direction. A 1°C increase in the NINO3.4 index through positive ENSO–temperature teleconnections was associated with an averagely 48.0% increase in annual dengue cases (95% CI = 9.5%–140.5%) in the occurrence year. Simultaneously, a 1°C increase in the NINO3.4 index through either positive or negative ENSO–precipitation teleconnections was associated with an averagely 12.2% increase (95% CI = 1.0%–27.2%) or 13.1% decrease (95% CI = 1.3%–26.0%) in annual dengue cases, respectively (Fig. 2c)."*

Comment 2.2:

El Niño impacts the local climate/weather on the African continent which has significant dengue transmission too. But this is not included in this study at all and the reasons for this/implications need at least discussion.

Response 2.2: We greatly appreciate this valuable observation regarding Africa's exclusion. We acknowledge that the effects of ENSO on dengue transmission in Africa can be significant. However, the long-term dengue case report data is currently not

available for African countries (Clarke, J. et al. 2024). We have now added this data limitation in the Discussion section.

Reference:

Clarke, J. et al. A global dataset of publicly available dengue case count data. *Sci. Data* 11, 296 (2024).

Lines 444–446 in Discussion (New): *“Forth, our exclusion of Africa due to surveillance disparities may impact the understanding of global ENSO-dengue mechanisms. Given Africa's substantial dengue burden, developing surveillance networks is needed to establish complete global mechanisms.”*

Comment 2.3:

Title

The authors focus solely on El Nino, maybe this should be reflected in the title?.

Response 2.3: Thanks for your suggestion. We have modified the title to “Rising dengue risk with increasing ENSO amplitude and teleconnections from 1980 to 2024”.

Comment 2.4:

Intro

I do not feel the term "teleconnections" is commonly used - it might be best to give a brief description at the beginning.

l.75 missing a word?

Response 2.4: Thanks for your concern. We have now added the concept and definition of “teleconnections” in the Introduction section and corrected the name of climate modes on line 75.

Lines 77–88 in Introduction (Revised): *“Given the extensive body of research, we hereafter use El Niño - Southern Oscillation (ENSO) as the well-known indicator of global climate variability, among other similar indices such as the Indian Ocean Dipole (IOD) and the Madden Julian Oscillation (MJO), to investigate this question for simplification.*

Manifested as sea surface temperature anomalies in the tropical Pacific Ocean, ENSO influences dengue risk through “teleconnections” – ENSO-induced atmospheric circulation alters local climates that affect mosquito ecology^{25,26}. Although the former process has been thoroughly investigated in climatology as teleconnections which is defined as the correlation between ENSO and local climate^{27,28}, an open question remains regarding the latter process; namely, how do teleconnections contribute to the different associations between ENSO and dengue epidemics²⁹?”

Comment 2.5:

Methods

eq 1: typo,  =

CMIP model selection - how do you define skilfull?

Response 2.5: Thanks for noticing the typo and skilfull model selection. We have now corrected the equation and extended the definition of skilfull models in the Methods section.

Line 516 in Methods (Revised):

$$PrEnv_i = \frac{\sum_{g=1}^n N_g \times Env_g}{\sum_{g=1}^n N_g} \quad (1)$$

Lines 677–689 in Methods (Revised): *“We measured ENSO amplitude as the standard deviation of the NINO3.4 time series⁴⁶. We also assessed the frequency of ENSO events but found a high uncertainty with great variations across multi-model ensemble members and a large bias between simulated and observed frequencies of ENSO events during 1980–2019. Thus, we chose the relatively robust ENSO amplitude to predict dengue cases.”*

“To ensure that our projections are feasible, we defined “skilful” ensemble members as the absolute value of the bias between simulated and observed ENSO amplitude during 1980–2019 is less than 50% of the observed ENSO amplitude⁵¹. Based on those selected skilful ensemble members, we calculated the population-weighted monthly temperature and daily precipitation to assess the teleconnections for each country during 1940–2019 and 2020–2099.”

Comment 2.6:

Results

Fig 1b is not easy to interpret - how do you explain this linear trend?

I.204 - why "mainly"? trade-off and synergy covers 100% of the area. It might be best to describe this figure in more detail.

Response 2.6: Thanks for noticing this issue. First, the main message of Fig. 1b is that more cases occurred during El Niño years than La Niña years, as most endemic countries are distributed above the one-to-one line. The linear trend between cases during El Niño and La Niña indicates the same message by showing that endemic countries with more cases during El Niño along with less cases during La Niña, which means that dengue cases in endemic countries are mainly affected by El Niño. We have now added the interpretation of the linear trend in the Results section. Second, we agree that trade-off and synergy cover 100% of the area. We have now removed “mainly” and extended the description of Fig. 2d.

Lines 126–132 in Results (Revised): *“Although long-term correlations between the time series of ENSO indices and annual dengue cases are insignificant in most countries, there exists a great magnitude of dengue epidemics in endemic countries such as Brazil, Vietnam, and Indonesia strongly affected by El Niño, i.e., more cases during El Niño years but fewer cases during La Niña years, as evidenced by both the linear trend and the clustering of endemic countries above the one-to-one line in Fig. 1b.”*

Lines 210–215 in Results (Revised): *“The aforementioned analysis suggests that ENSO teleconnections are persistently associated with dengue epidemics in two types via temperature and precipitation: trade-off (opposite effects via temperature and precipitation) and synergy (aligned effects via temperature and precipitation; Fig. 2d). Most countries (39/57) experienced trade-off in ENSO’s impacts on denque cases.”*

General Comments:

I briefly checked the Python code which seems to be doing what it should do. The full required data needs to be requested/downloaded from other sources but that is understandable.

There is a README with enough instructions.

Response to General Comments: Thank you for reviewing the code and documentation. We appreciate your confirmation of the code functions.

Reviewer 3:

Please note: The reviewer's comments are presented in regular text. The responses are in blue. Page and line numbers refer to those in the cleaned manuscript. Revised sentences in the updated manuscript are in *italics* and new sentences are underlined, presented at the end of each corresponding response.

General comments:

Dear Editors and Authors, thank you for the opportunity to review this study that explored the impact of ENSO on dengue fever. The manuscript is well drafted. Some comments for the authors to consider as attached. Hopefully, it can be helpful for further improvement.

Response to General Comments: We express our gratitude to the reviewer for providing positive comments and insightful suggestions on our manuscript. We have addressed the comments point by point. Please see our specific responses below.

Comment 3.1:

Introduction:

Line 81: It would be beneficial to further elaborate on the concept of teleconnection: What does it entail, and how does it contribute to the association between ENSO and dengue epidemics/incidence?

Response 3.1: Thanks for your great suggestion. We have now extended the concept and definition of “teleconnections” in the Introduction section.

Lines 82–88 in Introduction (Revised): *“Manifested as sea surface temperature anomalies in the tropical Pacific Ocean, ENSO influences dengue risk through “teleconnections” – ENSO-induced atmospheric circulation alters local climates that affect mosquito ecology^{25,26}. Although the former process has been thoroughly investigated in climatology as teleconnections which is defined as the correlation between ENSO and local climate^{27,28}, an open question remains regarding the latter process; namely, how do teleconnections contribute to the different associations between ENSO and dengue epidemics²⁹?”*

Comment 3.2:

Methods:

Line 459: This line states that data for the year 2024 were unavailable, and line 461 notes that data from 2020 to 2024 were not included. Does this mean that only dengue data from 1980 to 2019 were used in the analysis?

Response 3.2: Thanks for your question. We used the data from 1980 to 2019 in the main analysis and included the data in 2023 and 2024 in the sensitivity analysis (Supplementary Fig. 8c–d) to demonstrate the robustness of the main model. In line 475, we stated that data for the year 2024 were unavailable for four countries (Anguilla, Bhutan, Dominica, and Venezuela). We have now clarified this in the Methods section.

Lines 485–487 in Methods (Revised): *“Note that data from 2020 to 2024 were not included in the main model training to avoid the impact of COVID-19. We included the recent years 2023 and 2024 in the sensitivity analysis to demonstrate the robustness of our main model.”*

Comment 3.3:

Line 729: This line indicates that dengue data for 2023 and 2024 are available upon reasonable request. If these data were excluded from the analysis, what is the rationale for mentioning their availability? Please clarify.

Response 3.3: Thanks for your concern. We collected reported dengue case data for 2023 and 2024 and included them in the sensitivity analysis. We have now clarified this in the Data availability.

Lines 759–761 in Data availability (Revised): *“Reported dengue cases for 2023 and 2024 in sensitivity analysis are available upon reasonable request to the corresponding author and with permission from the data provider (Huaiyu Tian and Oliver J. Brady).”*

Comment 3.4:

Line 257 (Figure 3): Regarding the legend in Figure 3a: How can the reader identify areas marked as "not significant"? Are these limited to the small area along the China-India border? Please clarify.

Response 3.4: Thanks for noticing the nonsignificant countries. The effects of ENSO are not significant in Bhutan and some island countries. We have now listed these non-significant countries in the caption of Figure 3.

Lines 271–272 in Results (New): *“The effects of ENSO are not significant in Bhutan, New Caledonia, Bahamas, and Bermuda.”*

Comment 3.5:

Results:

Line 248: The manuscript states that the greater effects of the 2023-2024 events were likely due to a strong El Niño. However, since the 2023-2024 dengue data were excluded from the modeling, does this statement refer to a simulated effect or is it based on other sources of evidence? Please clarify.

Response 3.5: Thanks for raising this important question. Although the 2023–2024 dengue data were excluded from our main model, we demonstrated the robustness of the main model to including the 2023–2024 dengue data in our sensitivity analysis (Supplementary Fig. 8c–d). As such, we estimated the effects of the 2023–24 El Niño event on the number of dengue cases based on our main model. We have now clarified this in the Results section.

Lines 254–257 in Results (Revised): *“Since the effects are robust when including the data for 2023 and 2024 (Supplementary Fig. 8c–d), the recent 2023–24 El Niño event*

was predicted to induce an additional 9.6 million cases (95% CI = 6.3 to 12.6) in the occurrence year, with potentially more cases in 2025 and 2026.”

Reviewer 1:

Please note: The reviewer's comments are presented in regular text. The responses are in blue. Page and line numbers refer to those in the cleaned manuscript. Revised sentences in the updated manuscript are in *italics* and new sentences are underlined, presented at the end of each corresponding response.

General Comments:

I find that the authors have addressed my previous comments satisfactorily, and the manuscript is now considerably clearer and more coherent, with many of my concerns appropriately addressed.

However, I would like to point out a conceptual inaccuracy in the definition of teleconnections in Response 1.6.5, where it is stated that "... the former process has been thoroughly investigated in climatology as teleconnections which is defined as the correlation between ENSO and local climate," which is not entirely correct. In climatology, teleconnections are not merely statistical correlations, but rather physical atmospheric mechanisms that allow climatic anomalies in one region to influence distant areas. A well-documented example is the impact of ENSO on the Brazilian climate through shifts in the Walker circulation and the propagation of Rossby waves. I recommend rephrasing this section to better reflect the physical and dynamical basis of teleconnections.

Aside from this point, all my other concerns have been properly addressed, including the issues related to the figures, which are now more intuitive and clearer than before.

Response to General Comments: We appreciate the reviewer's positive feedback and the suggestions on the definition of teleconnections. We have now modified the definition to emphasize the physical and dynamical basis of teleconnections.

Lines 82–90 in Introduction (Revised): *"Manifested as sea surface temperature anomalies in the tropical Pacific Ocean, ENSO influences dengue risk through "teleconnections" – ENSO-induced atmospheric circulation such as the Walker circulation and the propagation of Rossby waves physically alter distant regional climates, thereby affecting mosquito ecology^{25,26}. While the dynamical mechanisms of these teleconnections have been well established in climatology^{27,28}, an open question remains regarding the latter process; namely, how do teleconnections contribute to the different associations between ENSO and dengue epidemics²⁹?"*

Reviewer 2:

General comments:

The authors have addressed all the reviewer's comments satisfyingly. I have got nothing further to add.

Response to General Comments: We appreciate the reviewer's positive feedback and thorough review of both the original and revised versions of our study.

Reviewer 3:**General comments:**

Dear Editors and Authors, thank you for the opportunity to review the revised manuscript. My concerns have been well addressed. Congratulations to the authors!

Response to General Comments: We appreciate the reviewer's positive feedback and thorough review of both the original and revised versions of our study.